# CHIT1 at diagnosis predicts faster disability progression and reflects early microglial activation in multiple sclerosis

Jarne Beliën [1,7], Stijn Swinnen [1,2,7], Robbe D'hondt [3,4], Laia Verdú de Juan[5], Nina Dedoncker[1], Patrick Matthys [6], Jan Bauer [5], Celine Vens [3,4], Sinéad Moylett[1,8] & Bénédicte Dubois [1,2,8] ✉

Multiple sclerosis (MS) is characterized by heterogeneity in disease course and prediction of long-term outcome remains a major challenge. Here, we investigate five myeloid markers – CHIT1, CHI3L1, sTREM2, GPNMB and CCL18 – in the cerebrospinal fluid (CSF) at diagnostic lumbar puncture in a longitudinal cohort of 192 MS patients. Through mixed-effects and machine learning models, we show that CHIT1 is a robust predictor for faster disability progression. Integrative analysis of 11 CSF and 26 central nervous system (CNS) parenchyma single-cell/nucleus RNA sequencing samples reveals *CHIT1* to be predominantly expressed by microglia located in active MS lesions and enriched for lipid metabolism pathways. Furthermore, we find *CHIT1* expression to accompany the transition from a homeostatic towards a more activated, MS-associated cell state in microglia. Neuropathological evaluation in *post-mortem* tissue from 12 MS patients confirms CHIT1 production by lipid-laden phagocytes in actively demyelinating lesions, already in early disease stages. Altogether, we provide a rationale for CHIT1 as an early biomarker for faster disability progression in MS.

Multiple sclerosis (MS) is a chronic autoimmune disorder of the central nervous system (CNS) characterized by considerable interindividual heterogeneity in terms of disease onset and severity[1]. Despite continuous scientific advancement, accurately predicting long-term outcomes and effectively stratifying treatment regimens based on uncertain prognosis remain major challenges in MS care. Clinicians thus face an unmet need for disease activity measures with proven prognostic value. In particular, very few blood and cerebrospinal fluid (CSF) biomarkers have yet found translation into clinical practice[2].

Over the past few years, the role of innate immune cells such as CNS-resident microglia and monocyte-derived macrophages in MS pathogenesis has gained increased attention[3–5]. Microglia are anticipated to provide novel strategies in the development of biomarkers for MS disease activity[6]. Previous work from our group showed that among several myeloid markers, CSF concentrations of chitotriosidase or chitinase 1 (CHIT1) at diagnostic lumbar puncture best reflected disability status at a median of five years later[7]. Only few other studies have evaluated CHIT1 concentrations in relation to MS prognosis[8,9]. However, these studies – as our previous work – were limited to single-time-point disability assessments. In addition, *CHIT1* has been found to be upregulated in chronic active MS lesions at the bulk RNA level, whereas its expression was largely undetectable in chronic inactive lesions and in brain tissue from non-neurological controls[7,10]. Nevertheless, since variations in biomarker concentrations

[1]Laboratory for Neuroimmunology, Department of Neurosciences, Leuven Brain Institute, KU Leuven, Leuven, Belgium. [2]Department of Neurology, University Hospitals Leuven, Leuven, Belgium. [3]Department of Public Health and Primary Care, KU Leuven, Kortrijk, Belgium. [4]Imec research group itec, KU Leuven, Kortrijk, Belgium. [5]Department of Neuroimmunology, Center for Brain Research, Medical University of Vienna, Vienna, Austria. [6]Laboratory of Immunobiology, Department of Microbiology, Immunology and Transplantation, Rega Institute for Medical Research, KU Leuven, Leuven, Belgium. [7]These authors contributed equally: Jarne Beliën, Stijn Swinnen. [8]These authors jointly supervised this work: Sinéad Moylett, Bénédicte Dubois. ✉e-mail: benedicte.dubois@uzleuven.be

must ultimately be coupled to disease processes to be clinically relevant, unraveling cell-type specificity of CHIT1 and the functional state of *CHIT1*-expressing cells in MS remains a prerequisite[11]. This is emphasized by the functional diversity and cellular heterogeneity of microglia and CNS-associated macrophages uncovered in recent work[12–14].

In this work, we investigate five microglia/macrophage-related proteins – CHIT1, chitinase-3-like protein 1 (CHI3L1, also known as YKL-40), soluble triggering receptor expressed on myeloid cells 2 (sTREM2), glycoprotein non-metastatic melanoma protein B (GPNMB) and C-C motif chemokine ligand 18 (CCL18) in the CSF at diagnostic lumbar puncture in a large cohort of 192 MS patients. Our longitudinal study design allows for multi-time-point disability assessments. Using mixed-effects models and machine learning algorithms, we identify CHIT1 at diagnosis as a strong predictor for faster disability progression. We consequently perform single-cell RNA sequencing (scRNA-seq) on the CSF of 11 MS patients and integrate this data with single-cell profiles from four published datasets of MS CNS tissue to elucidate the phenotype and localization of *CHIT1*-expressing cells. Lastly, we validate our results at the protein level using immunohistochemistry on *post-mortem* MS brain tissue in early as well as late disease stages.

## Results

### Study population for biomarker analysis

We investigated the association between five microglia/macrophage biomarkers in the CSF at diagnostic lumbar puncture and clinical disease activity in our longitudinal MS cohort. These five biomarkers were selected based on their role in myeloid activation and their potential diagnostic or prognostic value, as indicated in prior research on MS and other neurological disorders[10,15–17]. In this study, we included retrospective data of 196 MS patients. In contrast to our previous work in which we correlated CHIT1 concentrations in a subset of these patients ($n = 143$) with single-time-point disability assessments[7], we now examined additional biomarkers and included multi-time-point longitudinal disability outcomes in the extended MS cohort. None of these patients were on disease-modifying therapy at the time of CSF sample collection. After quality control of the CSF protein measurements, samples from 192 MS patients remained for analysis (Table 1, Supplementary Table 1, Supplementary Data 1). Of these 192 MS patients, we had disability assessments for 178 patients and information on relapses for 181 patients. Our study cohort was representative of the general MS patient population; 60.9% of patients were female, 80.7% of patients

presented with relapsing-remitting MS (RRMS) and the median age at onset was 33.9 years. The median disease duration at diagnostic lumbar puncture was 0.9 years.

### CHIT1 at diagnosis correlates with disability years later: single-time-point analysis

We observed some significant correlations between the five selected CSF biomarkers (CHIT1, CHI3L1, sTREM2, GPNMB and CCL18) (Supplementary Table 2), which is not unexpected since these are all implicated in microglia/macrophage biology. However, as not every correlation was significant, nor very strong, these biomarkers might reflect different aspects of microglia/macrophage biology and thus capture slightly distinct aspects of the disease processes in MS. Furthermore, CHIT1, CHI3L1 and GPNMB were associated with two of the CSF hallmarks currently used in MS clinical practice: immunoglobulin G (IgG) index and oligoclonal bands (OCBs; Supplementary Table 3). Notably, only CHIT1 correlated significantly with CSF neurofilament light chain (NfL; $r = 0.31$, $P < 0.0001$; Fig. 1A), an established marker of neuronal damage in MS[18].

Next, we examined how these microglia/macrophage CSF biomarkers measured at diagnosis correlated with single-time-point clinical disease activity parameters at follow-up years later. Disease parameters were assessed at a median of 5.4 years for disability and a median of 2 years for relapse activity before treatment. After correction for multiple testing, only CSF CHIT1 at diagnostic lumbar puncture showed a significant association with disability as measured by Age-Related Multiple Sclerosis Severity (ARMSS), Multiple Sclerosis Severity Score (MSSS) and Expanded Disability Status Scale (EDSS) 5.4 years later (Bonferroni-corrected $P \leq 0.0025$; Supplementary Table 4). Correlation of CHIT1 with disability was shown to be independent of CHI3L1 and NfL in our previous work[7]. Given the significant correlations found here between CHIT1, GPNMB and CCL18 (Supplementary Table 2), multiple linear regression analyses including these biomarkers were conducted for the disease activity outcomes. After Bonferroni correction for multiple testing, CHIT1 correlated with future disability independently of GPNMB and CCL18 (Supplementary Table 5). CSF CHIT1 concentrations explained 9.6% – and with known clinical covariates of disability (age at onset, sex and disease course) included 30.3% – of variance in ARMSS scores across MS patients. Analogous to our previous work, CHIT1 concentrations were significantly different between MS patients with high and low disability accumulation (ARMSS ≥ 5 and ARMSS < 5; logistic regression analysis,

## Table 1 | Characteristics of study population for biomarker analysis

| Characteristics | Subcohorts with measures of disease activity | | |
| --- | --- | --- | --- |
| | Total cohort<br>$n = 192$ | Disability<br>$n = 178$ | Relapse<br>$n = 181$ |
| Sex: female/male (% female) | 117/75 (60.9%) | 111/67 (62.4%) | 113/68 (62.4%) |
| Disease course: RRMS/PPMS/unknown (% RRMS) | 155/31/6 (80.7%) | 145/28/5 (81.5%) | 149/27/5 (82.3%) |
| Age at onset, years, median (IQR) | 33.9 (25.3–43.3) | 33.9 (25.1–43.1) | 33.8 (25.2–43.2) |
| Disease duration at LP, years, median (IQR) | 0.9 (0.1–5.1) | 0.8 (0.1–5.2) | 0.8 (0.1–5.1) |
| OCB status, positive/negative/unknown (% positive) | 168/22/2 (87.5%) | 155/21/2 (87.1%) | 158/21/2 (87.3%) |
| IgG index, median (IQR) | 0.9 (0.7 – 1.3) | 0.9 (0.7–1.3) | 0.9 (0.7–1.3) |
| Time from diagnosis to most recent EDSS, years, median (IQR) | – | 5.4 (2.6–8.7) | – |
| EDSS most recent, median (IQR) | – | 1.5 (1.0–3.5) | – |
| MSSS most recent, median (IQR) | – | 2.3 (0.9–4.7) | – |
| ARMSS most recent, median (IQR) | – | 2.2 (0.9–5.0) | – |
| Relapses from onset to treatment, median (IQR) | – | – | 2 (1–3) |
| ARR before treatment, median (IQR) | – | – | 0.6 (0.2–1.6) |

*ARMSS* Age-Related Multiple Sclerosis Severity, *ARR* annualized relapse rate, *EDSS* Expanded Disability Status Scale, *IgG* immunoglobulin G, *IQR* interquartile range, *LP* lumbar puncture, *MSSS* Multiple Sclerosis Severity Score, *OCB* oligoclonal bands, *PPMS* primary progressive MS, *RRMS* relapsing-remitting MS.

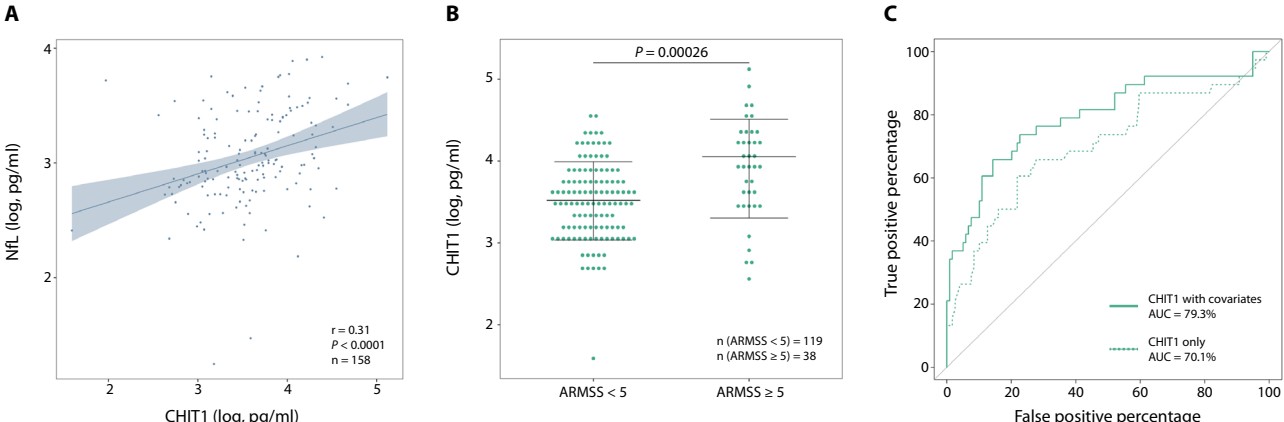

**Fig. 1 | CHIT1 at diagnosis correlates with disability years later: single-time-point analysis. A** Pairwise correlation (r) between CSF concentrations of CHIT1 *versus* NfL. *P*-value (*P*) is derived from two-sided Pearson correlation test. Shade indicates the 95% confidence interval around the linear regression line. **B** CSF CHIT1 concentrations at diagnostic lumbar puncture distinguish between MS patients with high and low disability accumulation (ARMSS ≥ 5 and ARMSS < 5). *P*-value (*P*) is derived from logistic regression analysis (generalized linear model) with CHIT1 as explanatory variable and high or low disability accumulation (ARMSS ≥ 5 and ARMSS < 5) as binary outcome variable after correction for clinical covariates of disability (age at onset, sex and disease course). Horizontal lines display the mean and standard deviation (SD). **C** Receiver operating characteristic (ROC) curves show the capacity to distinguish MS patients with high and low disability accumulation based on CSF CHIT1 concentrations alone or CHIT1 with clinical covariates included. AUC area under the curve. Source data are provided as a Source Data file.

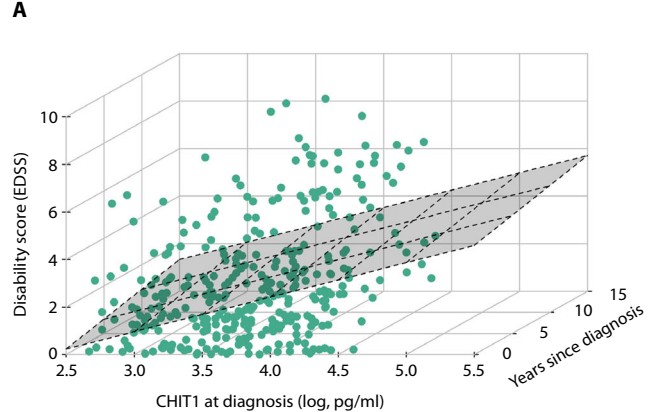

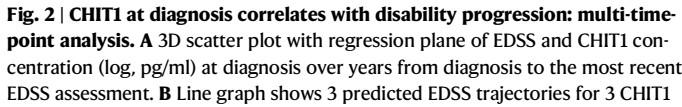

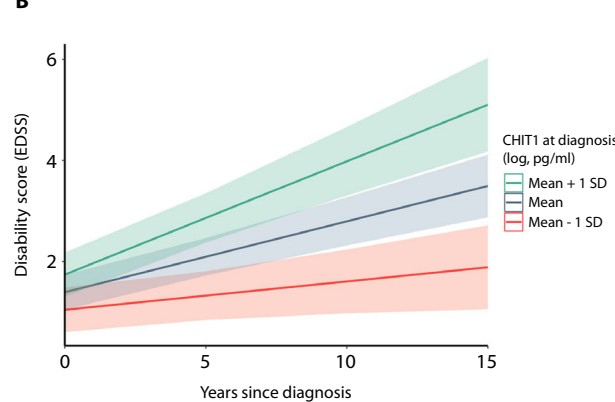

**Fig. 2 | CHIT1 at diagnosis correlates with disability progression: multi-time-point analysis. A** 3D scatter plot with regression plane of EDSS and CHIT1 concentration (log, pg/ml) at diagnosis over years from diagnosis to the most recent EDSS assessment. **B** Line graph shows 3 predicted EDSS trajectories for 3 CHIT1 concentrations (log, pg/ml) at diagnosis: the mean CHIT1 value in our cohort (3.63), the mean +1 SD (4.15) and the mean -1 SD (3.11), based on model 3. Shade indicates the 95% confidence intervals. SD standard deviation. Source data are provided as a Source Data file.

*P* = 0.00026; Fig. 1B) and the area under the curve (AUC) reached 70.1% (95% confidence interval [CI] = 59.4–80.7%) and 79.3% (95% CI = 69.8–88.7%) without and with known clinical covariates, respectively (Fig. 1C).

## CHIT1 at diagnosis correlates with disability progression: multi-time-point analysis

Given the above-mentioned associations between CSF CHIT1 concentrations at diagnostic lumbar puncture and single-time-point disability parameters, we expanded our investigations to multi-time-point EDSS assessments up to 15 years (median of 5.4 years) after MS diagnosis.

Likelihood ratio tests indicated that a mixed-effects model which accounted for EDSS variance across MS patients and across time – along with an interaction effect between CHIT1 and time – provided the best fit ($\chi^2$ = 9.94, *P* = 0.0016; model 3, Supplementary Table 6). Over 50% of MS patients (*n* = 104) had at least 2 EDSS assessments (Supplementary Table 7). In this model, MS patients with higher CSF

CHIT1 concentrations at diagnosis experienced more disability progression over time (β = 0.27, *P* = 2.63 ×10$^{-5}$; Fig. 2). As expected, there was a positive relation between EDSS and time (β = 0.20, *P* = 5.56 ×10$^{-8}$) as well as between EDSS and age at diagnosis (β = 0.42, *P* = 1.72 ×10$^{-11}$). There was no significant effect for sex, which was consistent across all models tested. Under random effects, we observed a positive correlation between by-patient random EDSS intercepts and by-time random EDSS slopes (r = 0.30). That is, MS patients with a higher EDSS at diagnosis tended to continue to have higher disability scores over time. After correction for fixed effects, we noticed that the interindividual EDSS differences accounted for a much greater proportion of the EDSS variance (77.8%) than the intraindividual disability progression over time (4.8%), as would be expected.

## After clinical prognostic factors, CHIT1 at diagnosis shows highest predictive value for disability progression

Next, we applied the machine learning approach we developed in D'hondt et al.[19] to model EDSS trajectories within our MS cohort. In

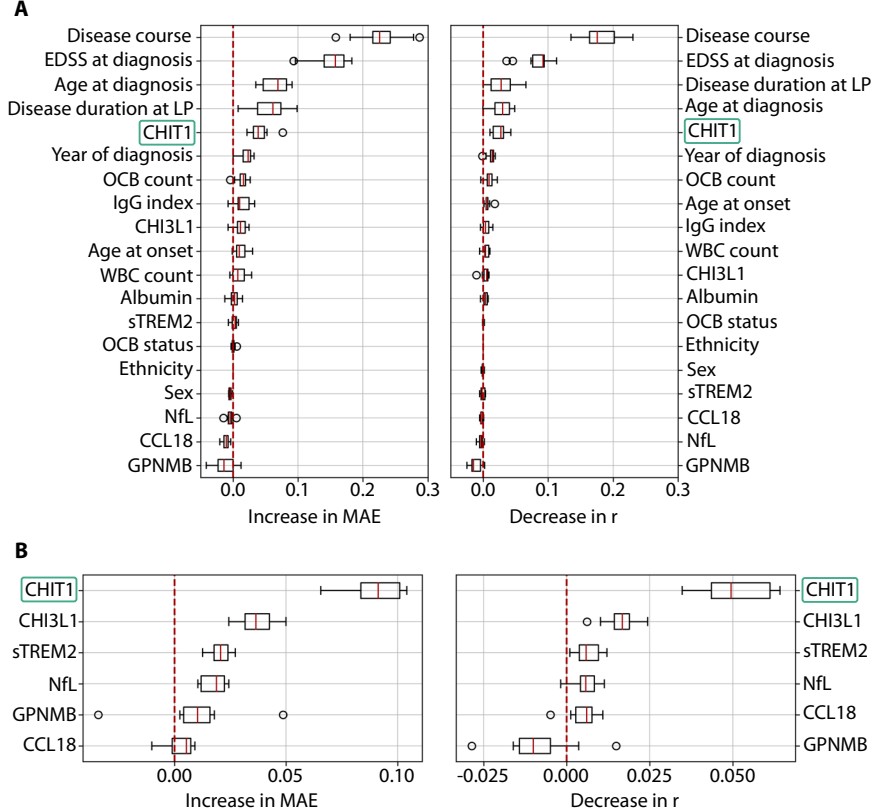

**Fig. 3 | After clinical prognostic factors, CHIT1 at diagnosis shows highest predictive value for disability progression. A** Permutation importance scores for each variable across 10 shuffles, showing the effect of randomly shuffling this variable on the mean absolute error (MAE) of the predictions (left) and on the Pearson correlation (r) between predictions and true values (right). The variables are ranked according to their average score. **B** Permutation importance scores for each CSF biomarker corrected for higher-order interactions, showing the effect of randomly shuffling any subset of biomarkers containing this biomarker on MAE (left) and r (right). Biomarkers in each relevant subset are independently shuffled and the effect on performance is divided by the subset size and the number of subsets of that size. This procedure is repeated 10 times for each biomarker. The variables are ranked according to their average score. Red line in box plots indicates the median importance for each variable. Bounds of box span the interquartile range (IQR) and whiskers indicate data up to 1.5× IQR from the bounds of the box. Values outside this range are considered outliers and are shown as circles. IgG immunoglobulin G, LP lumbar puncture, OCB oligoclonal bands, WBC white blood cells. Source data are provided as a Source Data file.

short, we used a regressor chain model to analyze the predictive power of several MS patient characteristics for disability progression, in particular the CSF biomarkers under study. To this end, we calculated permutation importance scores[20], which identify how much the random shuffle of a characteristic or variable affects the model's performance. As expected, we found disease course, baseline EDSS, age at diagnosis and disease duration at diagnosis to be important variables for the prediction of disability progression (Fig. 3A). However, after these, CSF CHIT1 concentrations at diagnosis emerged as the most important predictor. Our machine learning model thus demonstrated the prognostic superiority of CHIT1 over the other CSF biomarkers. This was even more apparent when higher-order CSF biomarker interactions were considered (Fig. 3B).

## CHIT1 is expressed by a distinct microglia subset in active MS lesions

In order to identify the cellular source of CHIT1 in the CSF of MS patients, we performed scRNA-seq on the CSF of 11 RRMS patients for whom we also had CSF CHIT1 measurements at diagnostic lumbar puncture. As our previous work showed *CHIT1* to be expressed in the brain of MS patients using bulk RNA analysis, we expanded our investigation by including scRNA-seq and single-nucleus (sn)RNA-seq datasets of CNS tissue from four published studies (Fig. 4A). Based on our earlier findings, we only included samples taken from lesions or periplaque white matter of MS patients[7]. With the exception of one patient,

all patients in the published datasets were diagnosed as progressive MS (either primary, secondary or undefined). For that one patient, the MS subtype was not further specified (Fig. 4A, Supplementary Data 2). All five datasets were subjected to stringent quality control. Ambient RNA contamination, doublets, genes present in less than five cells, cells with less than 200 genes detected and cells with a mitochondrial RNA content higher than 15% (for scRNA-seq data) or 5% (for snRNA-seq data) were all removed. In total, we retained 179,389 high-quality cells: 22,506 from CSF and 156,883 from CNS tissue (Supplementary Data 2). After confirming that *CHIT1* expression was indeed predominant in myeloid cells (Supplementary Fig. 1), we isolated a total of 29,676 myeloid cells: 1858 from CSF and 27,818 from CNS tissue. High-quality myeloid cells from all datasets were subsequently integrated, effectively correcting for batch effects and dataset-specific artefacts (Supplementary Fig. 2) prior to downstream analyses (Fig. 4A).

The myeloid cells were subclustered and annotation was performed based on canonical marker genes from the literature (Supplementary Data 3). We identified nine distinct microglia (MG1-9) clusters, two CNS-associated macrophage (CAM1-2) clusters and one monocyte (MON) cluster. One cluster portrayed expression of markers associated with both myeloid dendritic cells (mDC) as well as microglia and was termed 'mixed myeloid cells' (MMC; Fig. 4B). To confirm our annotation, we used unbiased module scores of gene sets for microglia, CNS-associated macrophages and monocytes from three recent reference works[12,21,22]. Most microglia clusters, and in particular MG4-7,

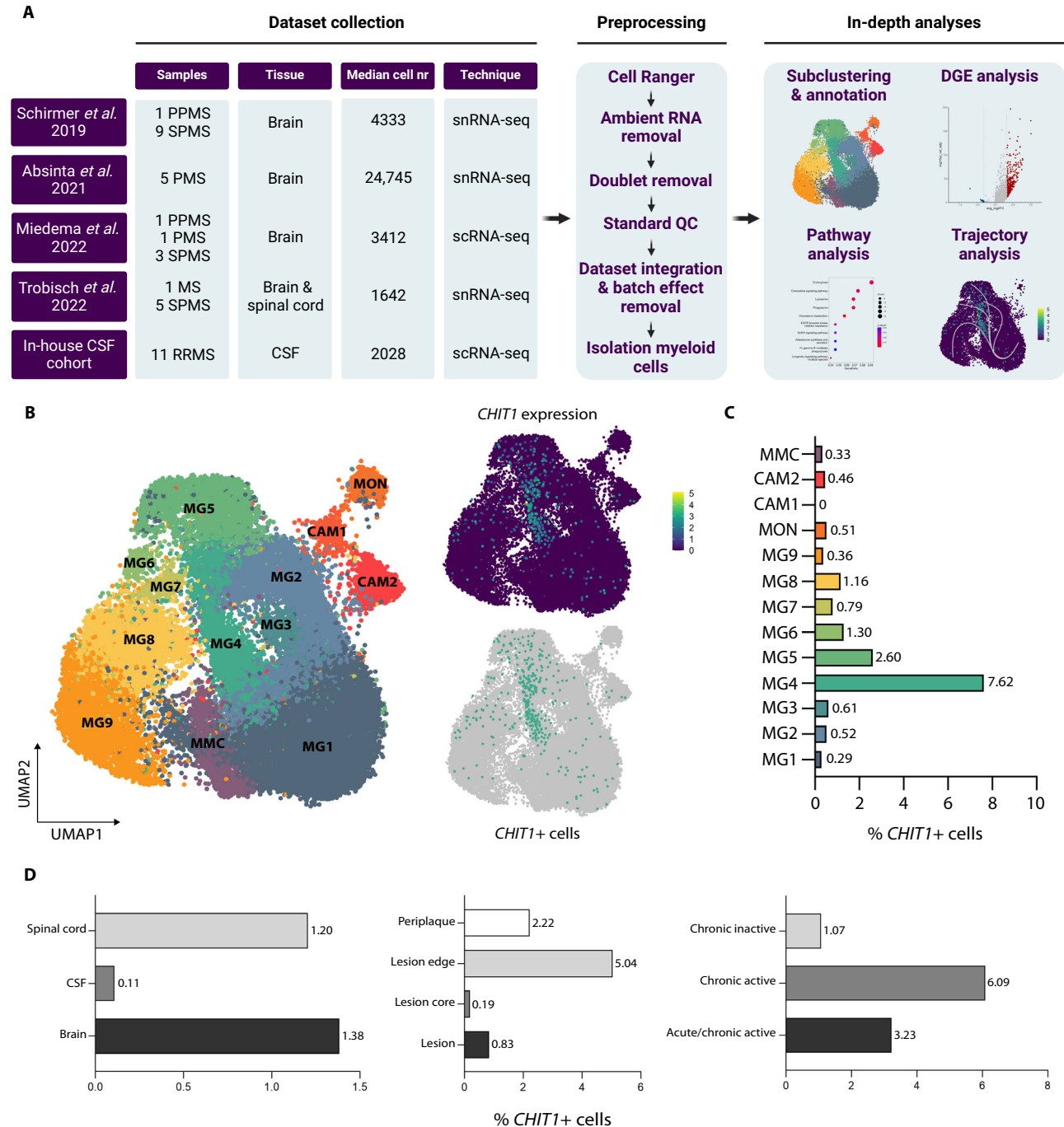

**Fig. 4 | *CHIT1* is expressed by a distinct microglia subset in active MS lesions.**
**A** Experimental setup and analysis of the transcriptomic data. **B** Uniform Manifold Approximation and Projection (UMAP) plots of the myeloid subclustering of the integrated dataset (left) show log-normalized *CHIT1* expression (top right) and *CHIT1*+ cells (bottom right). **C** Bar plot depicts the percentage of *CHIT1*+ cells per cluster. **D** Bar plots represent the percentage of *CHIT1*+ cells per tissue, localization within the lesion and lesion type. DGE differential gene expression, PMS progressive MS, PPMS primary progressive MS, QC quality control, RRMS relapsing-remitting MS, SPMS secondary progressive MS. Source data are provided as a Source Data file.

were enriched for the 'disease-associated microglia' (DAM) signature, which was expected as every sample in the different datasets was obtained from the disease-specific compartment (CSF and CNS) of MS patients (Supplementary Fig. 3A, B).

We found expression of *CHIT1* to be strongest in cluster MG4 (2625 cells; Fig. 4B) and more than half of *CHIT1*+ cells were indeed located in this cluster (200/386 *CHIT1*+ cells; Fig. 4C). *CHIT1*+ cells were almost exclusively found in CNS tissue (spinal cord and brain; Fig. 4D). Based on the metadata available for the CNS datasets, we found samples taken from the lesion edge as well as from chronic active lesions to

be most enriched for *CHIT1*+ cells (Fig. 4D). We verified that *CHIT1*+ cells were not a dataset-specific feature as all five datasets contributed to the total *CHIT1*+ cell pool (Supplementary Fig. 4). Module scoring of all *CHIT1*+ cells indicated that these cells portrayed a high transcriptional similarity to DAMs (Supplementary Fig. 3C).

**_CHIT1_+ microglia are associated with MS and foam cell differentiation**
Differential gene expression (DGE) analysis identified marker genes for each cluster (Fig. 5A). The top five enriched markers of cluster MG4

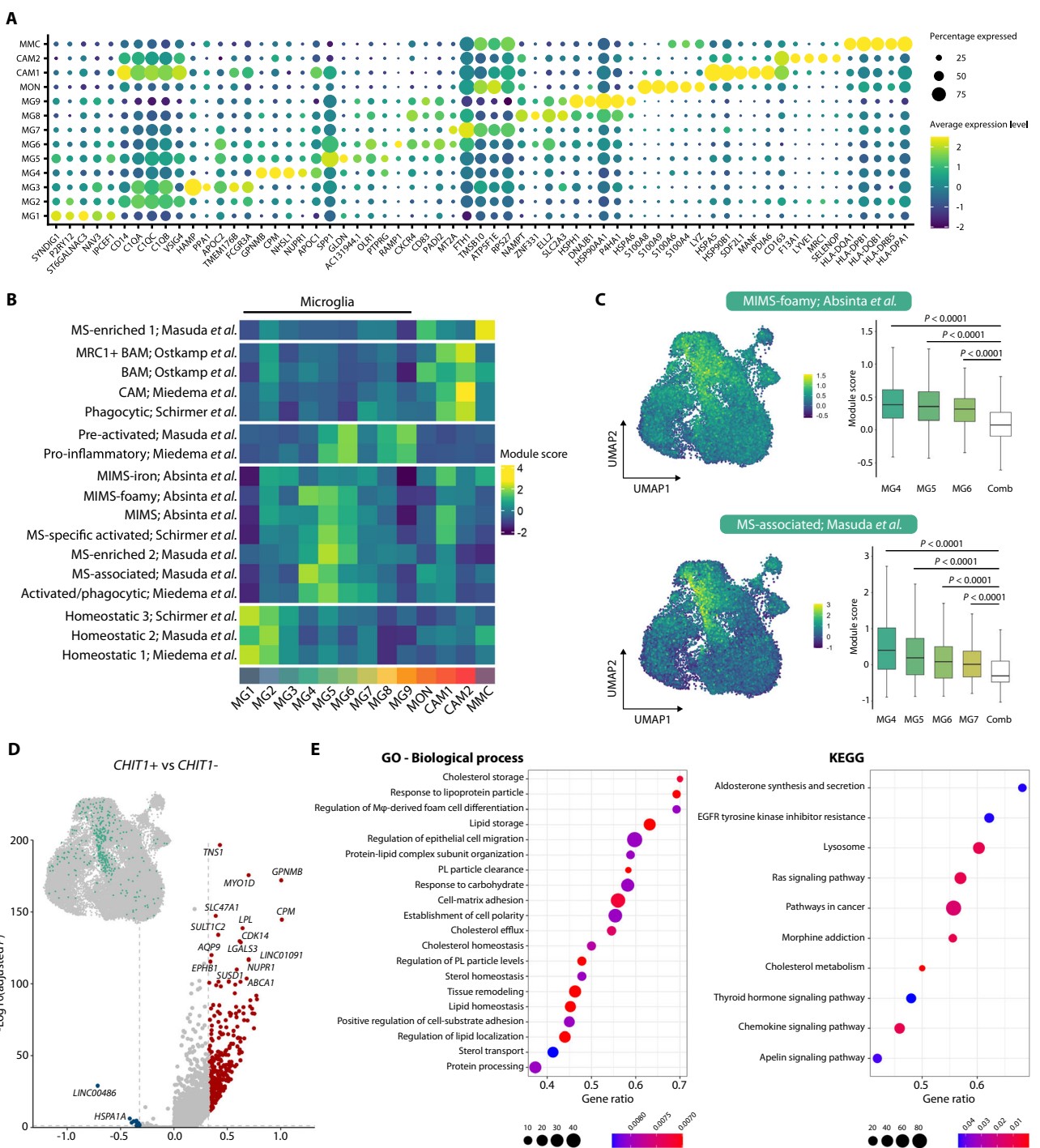

**Fig. 5 | *CHIT1*+ microglia are associated with MS and foam cell differentiation.**
**A** Dot plot shows the top five enriched genes for each cluster. Dot size reflects the percentage of cells within a cluster that express a particular gene. Color scale depicts the average log-normalized gene expression per cluster. **B** Heatmap shows the module scores per cluster for previously described microglia/macrophage subsets in MS[14,23–26]. **C** UMAP plots show the 'MIMS-foamy' (top) and 'MS-associated' (bottom) module score for all myeloid cells. Box plots depict median module scores over all cells per cluster where 'Comb' denotes combined median module score of all remaining clusters not shown as individual box plots. Bounds of box span the interquartile range (IQR) and whiskers indicate data up to 1.5× IQR from the bounds of the box. *P*-values (*P*) as defined by Kruskal-Wallis test followed by two-sided Dunn's test for pairwise multiple comparisons with Benjamini-Hochberg correction for multiple testing. **D** Volcano plot shows DGE between all *CHIT1*+ *versus CHIT1*- cells. Colored dots indicate significantly differentially expressed genes as determined by two-sided unpaired Wilcoxon rank-sum test (Bonferroni-corrected *P* ≤ 0.05) with a minimal increase (red) or decrease (blue) in expression of 25% [log2(fold change) = 0.322]. **E** Dot plots show significantly enriched pathways ranked by gene ratio. Dot size reflects the number of genes matched with the pathway gene set. Color of the dots depicts the Benjamini-Hochberg-corrected *P* from a one-sided Fisher's exact test. BAM border-associated macrophages, GO Gene Ontology, KEGG Kyoto Encyclopedia of Genes and Genomes, Mφ macrophage, PL plasma lipoprotein. Source data are provided as a Source Data file.

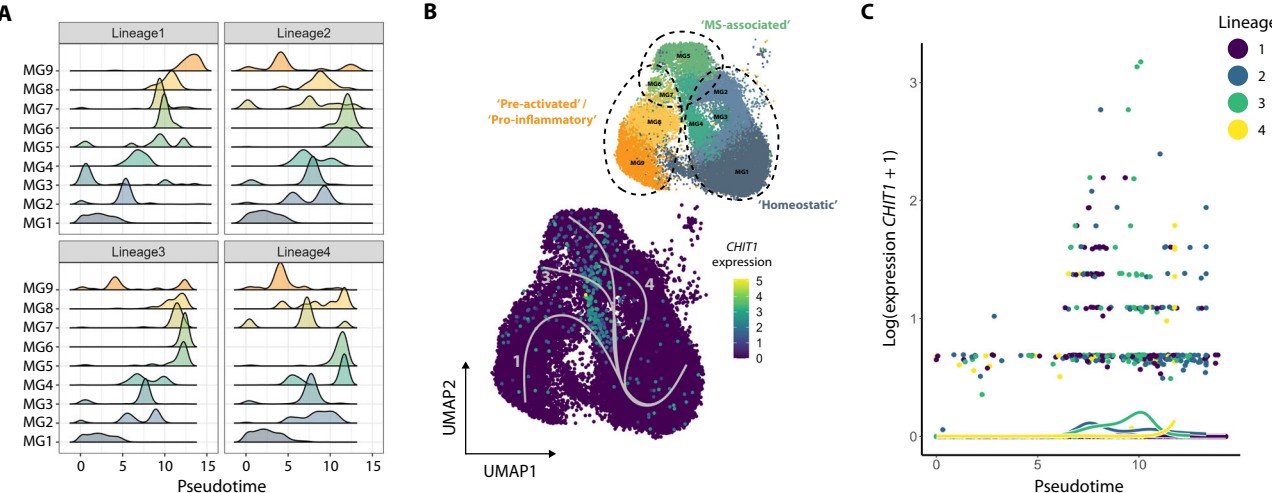

**Fig. 6 | *CHIT1* expression accompanies the transition from a homeostatic towards a more activated, MS-associated cell state in microglia. A** Ridge plot shows the density of cells for each cluster along the four lineages identified by pseudotime trajectory analysis. **B** UMAP plot shows the four trajectories running along the microglia clusters. Color scale depicts log-normalized *CHIT1* expression. Groups 'homeostatic', 'MS-associated' and 'pre-activated'/'pro-inflammatory' were based on the module score distribution in the previous part. **C** Scatter plot shows log-normalized expression of *CHIT1* along the trajectories. Dots depict counts of individual cells and lines reflect estimated smoothers for each lineage. Source data are provided as a Source Data file.

were *GPNMB*, *CPM*, *NHSL1*, *NUPR1* and *APOC1* (Supplementary Data 4). *CHIT1* did not come out as significantly upregulated in MG4 or in any other cluster. However, this is probably due to the overall low expression of *CHIT1* we observed in the data. Interestingly, *GPNMB* – one of the other biomarkers we measured in the CSF and which correlated significantly with CHIT1 – was the most differentially expressed gene in MG4 (1.91-fold change; Supplementary Data 4).

Next, we compiled gene modules for specific microglia/macrophage subsets previously described in MS[14,23–26] and once again calculated module scores (Fig. 5B). Overall, clusters MG1-3 were most enriched for 'homeostatic' signatures, MG4-7 for 'MS-associated', 'phagocytic' and 'MIMS-foamy' (or 'microglia inflamed in MS - foamy') signatures and MG8-9 for 'pre-activated' and 'pro-inflammatory' signatures. The two predominant modules for MG4 were 'MS-associated'[23] and 'MIMS-foamy'[25] (Fig. 5C). 'MS-associated' microglia were reported by Masuda and colleagues to express high levels of *CTSD*, *APOC1*, *GPNMB*, *ANXA2* and *LGALS1* and to be involved in de- and remyelination[23]. Absinta and colleagues found 'MIMS-foamy' to be enriched for pathways such as foam cell differentiation, lipid storage, response to lipoprotein particles and lysosome, which argues for a role in myelin phagocytosis and clearance[25].

To more specifically characterize *CHIT1+* cells, we performed DGE analysis of all *CHIT1+ versus CHIT1-* cells (Fig. 5D, Supplementary Data 5). Many of the genes upregulated in *CHIT1+* cells were found in the pathways as described for Absinta's 'MIMS-foamy' (e.g. *LPL*, *ABCA1* and *NUPR1*). Indeed, our own pathway analysis showed *CHIT1+* cells to be significantly enriched for several pathways related to foam cell differentiation, lipid homeostasis and clearance (Fig. 5E). Altogether, our analysis showed a considerable overlap in upregulated genes of cluster MG4 and *CHIT1+* cells with genes reported in 'MS-associated' and 'MIMS-foamy' signatures.

### *CHIT1* expression accompanies the transition from a homeostatic towards a more activated, MS-associated cell state in microglia

Based on our module scores, cluster MG4 – containing most of the *CHIT1+* cells – seemed to be positioned on the interface between homeostatic microglia and disease-/MS-associated microglia (Fig. 5B, Supplementary Fig. 3A, B). To investigate these apparent cell state dynamics within the microglia clusters, we performed pseudotime

trajectory analysis on clusters MG1-9 (Fig. 6, Supplementary Fig. 5). We identified four distinct lineages (Supplementary Fig. 5A), setting homeostatic cluster MG1 as the starting point. In lineages 1-3, cells belonging to cluster MG4 were positioned around the middle of the trajectories (at pseudotime ±7.5; Fig. 6A, Supplementary Fig. 5A). Global lineage structure showed MG4 to be an intermediate cluster, after which the trajectories diverge towards the 'MS-associated' clusters (MG5 and 6) on the one hand and the 'pre-activated'/'pro-inflammatory' clusters (MG8 and 9) on the other hand (Supplementary Fig. 5B). Indeed, when plotting the lineages, *CHIT1+* microglia were primarily located in-between the 'homeostatic' microglia at the start of the lineages and the 'MS-associated' and 'pre-activated'/'pro-inflammatory' microglia at the end of the lineages (Fig. 6B). Next, we performed DGE analysis over the course of the trajectories to validate this observation (Supplementary Data 6). *CHIT1* expression was found to be higher at the end point as compared to the starting point across all lineages (two-sided Wald test = 26.77, uncorrected $P = 2.21 \times 10^{-5}$). However, statistical significance did not hold after correction for multiple testing (Benjamini-Hochberg-corrected $P = 0.33$). Nevertheless, we did find *CHIT1* to be significantly enriched in cells positioned at pseudotime 7.5 (two-sided Wald test = 40.54, Benjamini-Hochberg-corrected $P = 5.01 \times 10^{-4}$). Moreover, *CHIT1* expression was significantly associated with pseudotime in lineage 3 (two-sided Wald test = 123.29, Benjamini-Hochberg-corrected $P = 5.92 \times 10^{-17}$; Fig. 6C, Supplementary Data 6) and peaked as pseudotime passed through cluster MG4 (Fig. 6). To sum up, pseudotime trajectory analysis indicated that *CHIT1* expression accompanies the transition of microglia towards a more activated, MS-associated cell state.

### Neuropathological evaluation shows phagocytic CHIT1+ cells in actively demyelinating lesions

To validate our transcriptomic results at the protein level, we performed neuropathological evaluation of brain tissue from 5 acute monophasic MS, 1 RRMS, 3 primary progressive (PPMS) and 3 secondary progressive (SPMS) MS patients. A total of 8 inactive and 21 active (including 2 chronic active or smoldering) MS lesions were analyzed (Supplementary Table 8).

Immunohistochemical staining of MS brains showed that CHIT1 was absent in normal-appearing white matter (NAWM, Supplementary Fig. 6). Where 95.2% of actively demyelinating lesions contained

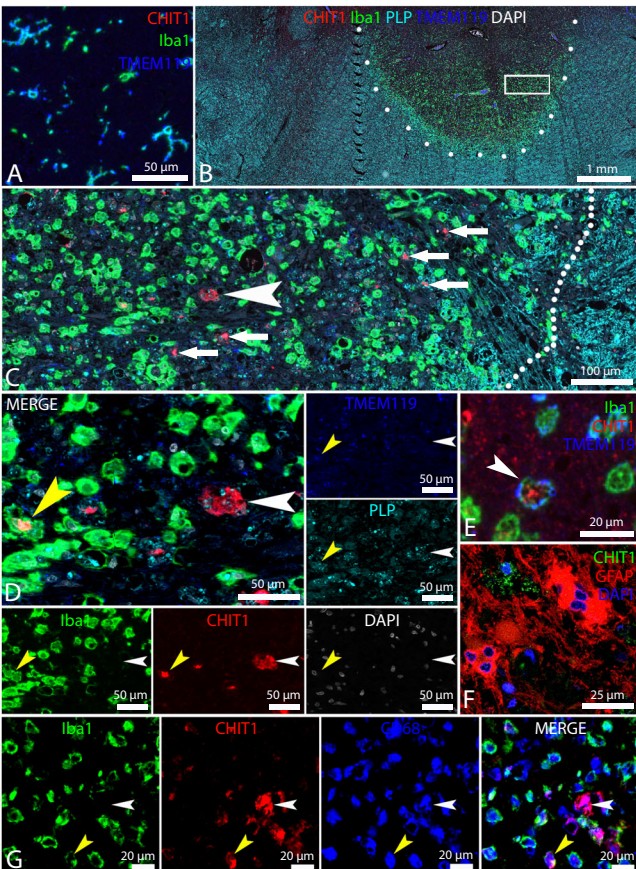

**Fig. 7 | Neuropathological evaluation shows phagocytic CHIT1+ cells in actively demyelinating lesions.** Representative images are shown of immunofluorescent multiplex staining in 7 active MS lesions from 4 individuals for CHIT1, Iba1 and TMEM119 in NAWM (**A**) and for CHIT1, Iba1, TMEM119, PLP, CD68 and DAPI (**B–G**). **A** Iba1+ TMEM119+ homeostatic microglia are CHIT1- in NAWM. **C** A magnification (rectangle) on the outer part of the active lesion in **B** is shown. The lesion edge is indicated by the dotted line. Demyelination is reflected by loss of staining for PLP. Small CHIT1+ cells are mostly found close to the lesion edge (small arrows). A larger CHIT1+ cell can be appreciated a bit further away from the lesion edge (arrowhead). **D** A further magnification of the same lesion shows a CHIT1+ Iba1+ cell (yellow arrowhead) and the larger CHIT1+ Iba1- cell (white arrowhead). Inside these cells, PLP+ degradation products reveal the phagocytic character of these cells. TMEM119 is absent on the larger CHIT1+ cell (white arrowhead). **E** Both TMEM119 and Iba1 are however present on small CHIT1+ cells (arrowhead). **F** Double staining for CHIT1 and GFAP demonstrates that CHIT1 is absent in astrocytes. **G** Triple staining for Iba1, CHIT1 and CD68 shows that CHIT1+ Iba1- cells are still phagocytic CD68+ cells (white arrowhead), just as CHIT1+ Iba1+ cells (yellow arrowhead). DAPI, 4′,6-Dia-midino-2-phenylindole.

CHIT1+ cells (20/21), none of the inactive lesions contained any CHIT1+ cells (0/8). We did not only observe CHIT1+ cells in active lesions of more late-stage MS patients (RRMS, PPMS and SPMS with a disease duration up to ±34 years), but also in acute (monophasic) MS patients (disease duration of 0.2-5 months) – confirming the presence of CHIT1 already early in the disease course. Of the two smoldering lesions, that only have an active lesion edge, one contained few CHIT1+ cells, whereas the other lacked CHIT1 immunoreactivity.

Immunofluorescent multiplex labeling confirmed that ionized calcium-binding adapter molecule 1-positive (Iba1+) and transmembrane protein 119-positive (TMEM119+) microglia do not express CHIT1 in NAWM (Fig. 7A). In active lesions however, small Iba1+ TMEM119+ cells with no or little uptake of proteolipid protein-positive (PLP+) degradation products were found to be CHIT1+ (Fig. 7B, C). Larger,

foamy (PLP-laden) CHIT1+ cells were mostly TMEM119- and some also Iba1- (Fig. 7C–E). This was in line with our transcriptomic analyses where we found that *AIF1* (the gene encoding for Iba1) was significantly downregulated in *CHIT1*+ cells as compared to *CHIT1*- cells (Supplementary Data 5) as well as along the pseudotime trajectories from homeostatic towards more activated, MS-associated microglia (Supplementary Data 6). Likewise, the loss of homeostatic microglia marker TMEM119 with the acquirement of CHIT1 expression corroborated our findings at the transcript level that CHIT1 expression in microglia accompanies the transition into a disease-/MS-associated phenotype. Interestingly, small CHIT1+ Iba1+ TMEM119+ cells (Fig. 7E) seemed to be present more towards the lesion edge, whereas larger, foamy CHIT1+ Iba1- TMEM119- cells were mostly found further away from the lesion edge (Fig. 7B, C). Double staining for CHIT1 and glial fibrillary acidic protein (GFAP) confirmed that none of the CHIT1+ cells were GFAP+ astrocytes (Fig. 7F). In addition, triple staining for CHIT1 together with Iba1 and CD68 illustrated that CHIT1+ Iba1- cells were still phagocytic CD68+ cells (Fig. 7G).

## Discussion

We demonstrated that CHIT1 concentration in CSF at diagnosis is a robust predictor for faster disability progression in MS patients and reflects early microglial activation. In comparison to several other microglia/macrophage-related biomarkers, only CHIT1 was found to correlate with future disability in our single-time-point analysis. By means of multi-time-point EDSS assessments and mixed-effects models, we observed that MS patients with higher CSF CHIT1 concentrations at diagnosis experienced more disability progression over time. Our machine learning models showed that CHIT1 had the highest predictive power for faster disability progression, after established clinical prognostic factors such as baseline EDSS. The consistency of our results across methodologies emphasizes the robustness of CSF CHIT1 as a biomarker. Through the analysis of single-cell transcriptomic profiles from CSF as well as CNS tissue of MS patients, we found *CHIT1* to be primarily expressed by a distinct microglia subset (MG4). These *CHIT1*+ cells were most abundant within active MS lesions. We discovered that the *CHIT1*+ cell state was enriched for pathways related to foam cell differentiation, lipid homeostasis and clearance, which suggests a role in the de- and/or remyelination processes in MS. Moreover, our pseudotime trajectory analysis illustrated that *CHIT1* expression in microglia accompanies the transition from a homeostatic towards a more activated, MS-associated phenotype. Neuropathological evaluation of both early- and late-stage MS patients showed that CHIT1+ cells were indeed present in almost all actively demyelinating lesions, whereas completely absent in inactive lesions and normal-appearing white matter. CHIT1+ cells were consistently CD68+ and PLP+, confirming their involvement in phagocytosis of myelin degradation products.

CHIT1 has been investigated as a putative biomarker in several other neurological disorders[27,28]. The most extensive research has been conducted in amyotrophic lateral sclerosis (ALS), a prototypic neurodegenerative disease. In parallel to our results, CSF CHIT1 has repeatedly been shown to correlate with faster disease progression in ALS[16,29]. CHIT1 has also been implicated in lipid storage disorders such as Gaucher, Niemann-Pick and Fabry disease. Particularly in Gaucher disease, plasma CHIT1 has been commonly used in clinical practice for over a decade to monitor disease severity and effectiveness of treatment[30,31]. In these lysosomal disorders, CHIT1 was found to be a marker for specialized lipid-laden macrophages or foam cells[30,32,33]. Interestingly, CHIT1 was suggested to reflect a particular activation or differentiation state of such lipid-laden macrophages rather than their absolute cell number[34]. In accordance with these findings, our pathway and trajectory analyses as well as neuropathological evaluation implied that CHIT1 in MS also marks a subset of activated microglia enriched for processes related to foam cell differentiation and lipid storage. Of

note is that most of the studies on lipid storage disorders have looked at the enzymatic activity of CHIT1 rather than its concentration in plasma. In our study, we specifically chose to focus on the concentration of CHIT1 and the other biomarkers for two reasons. First, literature on enzymatic activity of CHIT1 in CSF or plasma of MS patients is inconsistent with regard to correlation with disease course[35–38]. Second, compared to the assay for assessing CHIT1 enzymatic activity, measuring CHIT1 concentration is relatively quick, easy and inexpensive, facilitating its use in routine clinical practice.

A well-known constraint for the use of CHIT1 as a biomarker is the occurrence of a frequent 24-bp duplication in the *CHIT1* gene which leads to reduced expression levels and enzymatic activity[39]. As around 35% of individuals of European ancestry carry the variant[40], we have previously shown that *CHIT1* genotype did not substantially alter the correlation between CSF CHIT1 concentrations and disability[7]. Similar results have been reported in the context of ALS[41]. This suggests that variation in CSF CHIT1 concentrations does not mechanistically drive disease activity, but rather mirrors the contribution of microglia to disease. Since homozygous 24-bp duplication carriers – about 6% of individuals of European ancestry – have no or near-absent CHIT1, recommendations in Gaucher disease propose complementary *CHIT1* genotyping in case of non-elevated CHIT1 levels[42,43].

CHIT1 is a chitinase with as natural substrate chitin, an N-acetyl-glucosamine polymer and a structural component of fungi, nematodes and arthropods[28]. As chitin is not expressed in humans, chitinases are produced as a line of defense against these chitin-containing pathogens. In addition, CHIT1 has a role in the homeostasis of the innate immune system, where its precise mechanism of action remains unresolved[34]. In Alzheimer's disease, chitin-like glucosamine polymers due to impaired glucose utilization have been found within β-amyloid plaques[44] and CHIT1 secretion has been implicated in the clearance of these chitin-like polymers[45]. Although glucose metabolism is elevated in inflammatory conditions, Sotgiu and colleagues did not detect such chitin-like deposits in MS. The authors proposed that CHIT1 secretion by microglia/macrophages in MS effectively counteracts the deposition of chitin-like compounds, whereas in severe Alzheimer's disease this clearance is more imbalanced[46]. Alternatively, CHIT1 production in the absence of chitin might also represent an ancient remnant of microglia/macrophage activation as genes for chitinases are highly evolutionary conserved[34,46].

One of the principal challenges for the development of glial biomarkers in MS has been cellular specificity. Several CSF and serum biomarkers of myeloid cell activity in MS are under investigation (e.g. CHI3L1, soluble CD163 and osteopontin). However, these biomarkers do not sufficiently distinguish between CNS-resident microglia and monocyte-derived macrophages, which complicates their applicability since microglia and macrophages might effectuate different roles in MS[11]. Here, our transcriptomics showed that *CHIT1* expression is largely restricted to microglia and accompanies their transition towards a more activated, MS-associated cell state. After the *CHIT1*+ cluster MG4, our microglial trajectories diverged towards the 'MS-associated' clusters (MG5 and 6) on the one hand and the 'pre-activated'/'pro-inflammatory' clusters (MG8 and 9) on the other hand. This trajectory pattern suggests that homeostatic microglia upregulate *CHIT1* when they start to get activated and involved in lipid clearance. However, *CHIT1* expression appeared to wane again in most microglia when they start to develop a disease-/MS-associated phenotype, possibly instigated by unrelenting inflammatory cues and/or persistent lipid phagocytosis and accumulation. The second direction towards 'pre-activated'/'pro-inflammatory' clusters might reflect microglia present in the MS brain that obtain a more general activated, inflammatory profile, possibly as a form of bystander activation, although more in-depth functional analyses of these cells are needed to confirm such hypotheses. Neuropathological evaluation showed that small CHIT1+ cells with limited PLP uptake were Iba1+ and TMEM119+ and were mostly situated towards the lesion edge, whereas this Iba1 and TMEM119 immunoreactivity was absent on larger, foamy CHIT1+ cells located more away from the lesion edge, suggesting that CHIT1+ microglia start to lose certain surface markers in the process of PLP accumulation. TMEM119 is one of the main homeostatic microglia markers that is not expressed on monocyte-derived macrophages and has been shown to be downregulated in active MS lesions and other inflammatory conditions of the brain[47–49]. Moreover, decreased Iba1 expression and even Iba1- microglia have previously been described in neurodegenerative disorders such as Huntington's and Alzheimer's disease[50]. Keren-Shaul and colleagues showed that DAMs found in a mouse model of Alzheimer's disease were characterized by downregulation of *AIF1* (the gene encoding for Iba1) as well as homeostatic genes such as *TMEM119* and *P2RY12*[51].

Although relapsing and progressive MS are classified as distinct clinical phenotypes, the field now increasingly recognizes that the clinical course of MS is a continuum, where both inflammatory and neurodegenerative processes co-exist in a spectrum[52]. Moreover, it is known that activation of CNS-resident microglia already occurs early in the disease[53]. Indeed, most phagocytes initially present in active MS lesions are microglia, and with lesion maturation, more macrophages will infiltrate from the blood[47]. As MS shifts towards a more progressive phase, microglia are once more involved in the slow expansion of chronic active lesions[54]. Our work suggests that already early in the MS disease course CHIT1 is secreted into the CNS parenchyma by activated microglia who lost their homeostatic signature as these cells start to accumulate myelin debris. Conceivably, interstitial CHIT1 consequently ends up in the CSF via the glia-lymphatic or 'glymphatic' system in which CSF mixes with the interstitial fluid to wash away extracellular waste, as has been described for β-amyloid[55,56]. Elevated CSF CHIT1 concentrations at diagnosis thus likely identify MS patients who already at the start of their disease present with significant CNS pathology – i.e. more extensive microglial activation – and are at greater risk for faster disability progression. As of now, apart from some clinical parameters such as baseline EDSS, older age at diagnosis and spinal cord lesions, MS clinicians are equipped with few predictors for disability progression at time of diagnosis[57]. Therefore, a fast-progressive disease course is now often only recognized retrospectively, when damage has already been acquired. In particular with the advent of CNS-penetrant therapies that modulate microglial activity such as Bruton's tyrosine kinase inhibitors[58], the use of microglia biomarkers such as CHIT1 to inform the benefit of these treatments might significantly improve MS outcome.

We note some limitations of our study. First, the clinical data in our MS cohort comes from a real-world clinical environment. Although real-world data achieves high external validity, it introduces non-standardized time-points for disability assessment. To mitigate this limitation, we included covariates that reflect biological differences where relevant and chose analytical methods suitable for unbalanced datasets. With regard to the transcriptomics, the absolute number of *CHIT1*+ cells was small and within these cells expression of *CHIT1* was relatively low, which might have hampered the power of downstream analyses. However, as reported, most statistical tests on *CHIT1*+ cells yielded highly significant results ($P \leq 0.001$) after correction for multiple testing. Furthermore, as we worked with publicly available datasets, some of the transcriptomic analyses were limited by the metadata available in these studies. In particular, evaluation of the enrichment of *CHIT1* transcripts within certain lesion types or lesion edge *versus* core might have been impeded by inconsistent labeling of samples across studies.

Further validation is indispensable before employment of CSF CHIT1 concentration as a prognostic biomarker in MS clinical practice. First and foremost, the prognostic value of CHIT1 should be re-evaluated in a large, prospective and independent MS cohort, which will also be instrumental to define clinically relevant thresholds of CSF

CHIT1 concentrations for patient stratification. Secondly, as our primary outcome for the mixed-effects and machine learning models was EDSS, we were not able to dissect the relative contribution of relapse-associated worsening to the global disability progression in our MS cohort. Although CSF CHIT1 concentration was not significantly correlated with annualized relapse rate in this study, an alternative primary endpoint to use in future studies to address this would be 'progression independent of relapse activity' or PIRA[59]. Lastly, as we here provided initial biological insights into the prognostic potential of CHIT1 in MS, now more in-depth neuropathology as well as functional cellular studies are needed to 1) further determine the exact location of CHIT1 cells within different lesion types and 2) infer their exact phenotype and role in the de- and/or remyelination processes in MS.

In summary, we combined various complementary methods to propose CSF CHIT1 concentration at diagnosis as a putative biomarker to predict faster disability progression and reflect early microglial activation in MS. Our work provides a rationale for further validation with a prospect to clinical implementation. We believe CHIT1 has potential to cater to the unmet need for tools to aid MS clinicians in patient stratification and therapy selection, which might enable a more personalized approach in the treatment of MS patients.

## Methods

All MS patients provided written informed consent for CSF sample collection and data analysis. This study was approved by the Ethics Committee of the University Hospitals Leuven (S60222 and S50354) and performed in accordance with the declaration of Helsinki. The use of *post-mortem* brain samples from the archives of the Center for Brain Research was approved by the Ethics Committee of the Medical University of Vienna (EK. Nr.: 535-2004).

### Study design

In this study, we aimed to identify a strong predictive biomarker for disability progression in MS amongst five microglia/macrophage-related proteins measured in the CSF at diagnostic lumbar puncture – CHIT1, CHI3L1, sTREM2, GPNMB and CCL18. To this end, we extended our previously described cohort ($n = 143$)[7] to 196 MS patients. Protein concentrations were measured in duplicate and blinded to clinical data. After rigorous quality control (QC), 192 MS patients were retained for further analyses. Primary and secondary endpoints were defined in advance. For single-time-point analysis, primary outcomes were ARMSS and annualized relapse rate (ARR) before treatment and secondary outcomes were MSSS and EDSS. For multi-time-point analysis (mixed-effects models and machine learning algorithms), we assessed EDSS as the only outcome. Next, we addressed the cellular source of CHIT1 – our most robust CSF biomarker – to provide a biological rationale for its prognostic value. We performed scRNA-seq on CSF of 11 MS patients and supplemented these data with single-cell profiles of MS CNS tissue from four publicly available sc/snRNA-seq datasets. Samples of interest were included based on prior knowledge as described below. All CSF and CNS transcriptomes that failed stringent QC were removed from downstream analyses. Finally, we corroborated our observed *CHIT1* expression patterns at the protein level with immunohistochemistry on *post-mortem* brain tissue of MS patients, both in early and late disease stages. Tissue selection was informed by results from transcriptomics.

### Biomarker analysis

**Study population.** All MS patients were diagnosed according to the 2017 revised McDonald criteria[60] at the Department of Neurology of the University Hospitals Leuven (Belgium). We distinguished disease course based on a relapsing-remitting or primary progressive MS onset (RRMS or PPMS). Clinical data were collected by the same expert neurologist (BD) at diagnosis and follow-up visits up to 15 years (median of 5.4 years) after diagnosis, effectively minimizing information bias. All EDSS assessments were done when MS patients were not in active relapse. To control for the effect of age and disease duration on our disability measures, individual EDSS scores were converted to ARMSS and MSSS values, respectively[61,62]. The ARR was calculated over a minimum period of 90 days before the start of treatment, as reported earlier[63]. During disability follow-up (EDSS, ARMSS and MSSS), 62% of MS patients were on disease-modifying therapies.

**CSF protein measurements.** CSF of MS patients was collected during diagnostic lumbar puncture as part of standard clinical care. None of these patients were on disease-modifying therapy at the time of sample collection. Polypropylene tubes were centrifuged for 10 min at $3000 \times g$ at 4 °C and CSF supernatant was stored at −80 °C until protein measurements. We assessed CHIT1, CHI3L1, sTREM2, GPNMB and CCL18. NfL was also included as canonical biomarker for neuronal damage. CHIT1, GPNMB, CCL18 and NfL were measured within the entire MS cohort ($n = 196$). For CHI3L1 and sTREM2, measurements were limited to the original cohort ($n = 143$) as previously described[7].

CSF protein concentrations were measured in duplicate and blinded to clinical data. For CHIT1, the CircuLex Human Chito-triosidase ELISA Kit (#CY-8074, MBL Life Science) was used. GPNMB and CCL18 concentrations were quantified with U-PLEX technology (#F21ZH-3 and #F212H-3, MSD). CHI3L1, sTREM2 and NfL measurements were obtained as previously described[7]. We performed all assays conforming to the manufacturer's instructions.

QC was carried out as follows: we checked for consistency of the standard curves across plates, with $0.74 < r < 1$ ($P \leq 0.05$) for all assays. Concerning plate-to-plate variability, we calculated the coefficient of variation (CV) based on an internal control sample measured on all plates. Samples with a CV more than 20% or CSF biomarker concentrations below the detection limit were excluded. Samples above the upper detection limit were remeasured using a different dilution factor where possible or were eliminated otherwise. Within-plate variability was assessed based on the CV across duplicates (Supplementary Table 1). Mean values across duplicates were used for analysis.

### Single-cell transcriptomics

**Datasets.** In-house CSF cohort: All patients ($n = 11$) were diagnosed with RRMS according to the 2017 revised McDonald criteria[60] by the same expert neurologist (BD) at the Department of Neurology of the University Hospitals Leuven (Belgium) (Supplementary Data 2). None of these patients were on disease-modifying therapy at the time of sample collection.

Previously published CNS datasets: All previously published CNS datasets were downloaded as FASTQ files from the Sequence Read Archive [SRA, National Center for Biotechnology Information (NCBI)] under BioProject accession numbers PRJNA544731 [Schirmer et al.[24]], PRJNA749443 [Absinta et al.[25]], PRJNA743676 [Miedema et al.[14]] and PRJNA726991 [Trobisch et al.[64]]. The data from Miedema et al.[14] was acquired via scRNA-seq, whereas the others performed snRNA-seq. From all studies, only samples of interest were used, i.e. CNS lesions or periplaque white matter of MS patients (Supplementary Data 2). CNS data was processed in parallel to our in-house CSF data from the Cell Ranger count pipeline onwards.

**CSF sample collection and sequencing.** CSF samples were obtained during diagnostic lumbar puncture. 5 ml of CSF was collected into round-bottom polypropylene tubes (Sarstedt) and processed within 1 hour after lumbar puncture to ensure optimal sample quality. CSF samples were centrifuged for 15 min at $400 \times g$ at 4 °C. We resuspended the CSF cell pellets in 100 μl of phosphate-buffered saline (PBS, Gibco) and counted the cells on a LUNA-FL Dual Fluorescence Cell Counter (Logos Biosystems). CSF cells were centrifuged again (15 min, $400 \times g$, 4 °C) and resuspended in 1.5 ml of RPMI 1640 Medium (Gibco)

containing 20% heat-inactivated fetal bovine serum (FBS, Merck) and 10% dimethyl sulfoxide (DMSO, Merck). CSF cells were transferred to Nunc CryoTubes (Thermo Scientific) and stored in a CoolCell container (Corning) at −80 °C. After 24 h, cells were relocated to liquid nitrogen for cryopreservation until further processing. Cryopreservation allowed us to limit batch effects without a substantial effect on gene expression patterns[65,66].

Library preparation and scRNA-seq of our in-house CSF cohort was performed in three separate batches. The average latency between cryopreservation and library construction was 4.3 months (Supplementary Data 2). For every batch, we resuscitated the cryopreserved CSF cells in warmed RPMI 1640 Medium (Gibco) containing 20% heat-inactivated FBS (Merck). Cells were centrifuged for 10 min at 400× $g$ at room temperature (RT) and resuspended in PBS (Gibco) containing 0.04% bovine serum albumin (BSA, Merck). We consistently obtained a high cell viability (mean: 94.7% ± 5.8% SD). Single-cell suspensions were loaded on a Chromium Next GEM Chip K for single-cell partitioning and barcoding with the Chromium Controller according to the Chromium Next GEM Single Cell 5′ Reagent Kit v2 (Dual Index) (#1000286 and #1000263, 10x Genomics). We performed further library preparation conforming to the manufacturer's instructions. Libraries were sequenced on an Illumina NovaSeq 6000 with a target sequencing depth of 50,000 reads per cell.

**Preprocessing of sequencing data.** Processing of the sequencing data from both our in-house CSF cohort as well as the four previously published CNS datasets was performed with Cell Ranger v7.1.0 software (10x Genomics). Read alignment and transcript count were done individually for each sample via the Cell Ranger count pipeline with default parameters. Human reference GRCh38 (GENCODE v32/Ensembl 98) version 2020-A (July, 2020) was used for gene mapping.

**Quality control.** Before integration, all samples from all CSF and CNS datasets went separately through our QC pipeline in R v4.2.2. Ambient RNA contamination was removed using the SoupX v1.6.2 package[67] with default parameters, except for samples from Miedema et al.[14] where tfidfMin was adjusted to 0.8 (default 1) due to the lower cellular complexity of the dataset (prior FACS). Doublets were identified via scDblFinder v1.12.0[68] with default parameters. The corrected count matrices and relevant metadata were used to create Seurat objects for further analysis with Seurat v4.3.0[69]. Features (genes) present in less than five cells were removed and only high-quality cells were retained, i.e. singlets with more than 200 features and a maximal percentage of mitochondrial RNA content of 15% for scRNA-seq data[70] and 5% for snRNA-seq data[25,64] (Supplementary Data 2). We normalized and scaled the count matrices via SCTransform v0.3.5[71] with default parameters. Mitochondrial and ribosomal content as well as the difference in expression between S and G2M cell cycle genes were regressed out.

**Data integration, batch effect removal and myeloid clustering.** In order to harmonize the different samples and avoid subject-specific clusters, we integrated the individual samples of the five datasets (our in-house CSF dataset and four previously published CNS datasets) into one dataset using the Harmony v0.1.1 package[72] with default parameters. We assessed different integration strategies with various (combinations of) covariates to correct for. Best integration was obtained when each sample was treated as a batch and sequencing technique (scRNA-seq or snRNA-seq) was included as a variable. To cluster the cells, FindNeighbours() was run with Harmony-corrected principal components, followed by FindClusters(). We visualized the clusters on a Uniform Manifold Approximation and Projection (UMAP) plot. Optimal cluster resolution was determined with guidance of the clustree v0.5.0 package[73]. To identify myeloid clusters, we checked the expression of canonical myeloid marker genes (Supplementary Data

3). Furthermore, we used SingleR v2.0.0[74] for automated annotation based on the Novershtern immune reference dataset[75] from the celldex v1.8.0 package[74]. Non-myeloid clusters were removed.

**Annotation of myeloid clustering.** DGE analysis was performed using FindAllmarkers() with default parameters, except for min.pct = 0.25 (default 0.10), to investigate the most differentially expressed genes (DEGs) in each cluster. We subdivided the myeloid clusters into monocytes, macrophages and microglia based on canonical marker genes (Supplementary Data 3). For more in-depth annotation of these clusters, we compiled gene sets of interest (or modules) from previously published articles (Supplementary Data 3) and scored their expression levels in our integrated dataset via AddModuleScore()[76].

**Pathway analysis.** *CHIT1*+ cells were identified via WhichCells() as cells with an expression value > 0 in the RNA assay. We used FindMarkers() to get DEGs between *CHIT1*+ and *CHIT1*- cells. Only genes with an adjusted *p*-value < 0.05 were retained and used as input for Gene Ontology (GO) and Kyoto Encyclopedia of Genes and Genomes (KEGG) gene set enrichment analysis via the clusterProfiler v4.6.2 package[77].

**Pseudotime trajectory analysis.** To estimate cell state dynamics, we performed pseudotime trajectory analysis using the slingshot v2.6.0[78] and tradeSeq v1.12.0[79] packages. Homeostatic cluster "Microglia 1" (MG1) was defined as the starting point. Since the inclusion of non-microglia clusters ["Monocytes" (MON), "CNS-associated macrophages 1 and 2" (CAM1 and CAM2) and "Mixed myeloid cells" (MMC)] might lead to nonsensical trajectories, we subsetted our data to only retain microglia clusters for pseudotime trajectory analysis to gauge where the *CHIT1*+ microglia were positioned among the different microglia cell states. Moreover, we carried out DGE analysis along the inferred trajectories with fitGAM(), assoRes() and startVsEndTest().

## Neuropathological evaluation

**Patients.** For neuropathological evaluation, formalin-fixed paraffin-embedded (FFPE) autopsy material from the archives of the Center for Brain Research (Medical University of Vienna) was used. We assessed samples from acute monophasic MS [*n* = 5, described before[80]], RRMS (*n* = 1), PPMS (*n* = 3) and SPMS (*n* = 3) patients. From these MS patients, we analyzed a total of 21 active (2 smoldering) and 8 inactive lesions (Supplementary Table 8).

**Luxol Fast Blue-Periodic acid Schiff staining for myelin.** 5 μm thick paraffin sections of MS brain were stained with Luxol Fast Blue-Periodic acid Schiff (LFB-PAS) according to standard procedure to detect demyelinated lesions and macrophages with myelin degradation products.

**Immune light microscopical staining.** 3-5 μm FFPE sections were deparaffinized in xylene (2× 15 min) and rinsed with 96% ethanol. To block endogenous peroxidase activity, samples were incubated in $H_2O_2$-methanol (30 min). Next, we gradually rehydrated the samples with ethanol (96%, 70%, 50%) and finally deionized water. Heat-induced epitope retrieval (HIER) was performed with ethylenediaminetetraacetic acid (EDTA, pH 9.0) in a household food steamer (Braun, 1 h). Afterwards, we rinsed the sections with tris-buffered saline solution (TBS) 3-5 times. Non-specific background reactions were blocked with 10% fetal calf serum (FCS) in DAKO buffer (Agilent) for 15 min. Hereafter, we incubated the slides with primary antibody CHIT1 (#HPA010575, Sigma, 1:100, 4 °C overnight) or CD68 (#M0814, Dakocytomation, 1:100, 4 °C overnight) and rinsed them with TBS before incubating them with a secondary biotin anti-rabbit-conjugated antibody (1:200, #711-065-152, Jackson, 1 h at RT) or biotin-anti-mouse-conjugated antibody (#715-065-150, Jackson, 1:500, 1 h at RT). Next, the slides were rinsed with TBS and incubated with peroxidase-conjugated

streptavidin (1:500) in 10% FCS-DAKO buffer (1 h at RT). Finally, the sections were rinsed with TBS, developed with 3,3'-Diaminobenzidine (DAB) and counterstained in hematoxylin (15-20 s) before dehydration and mounting with cover slips using EUKITT (ORSAtec).

**Immunofluorescent multiplex labeling.** Immunofluorescent multiplex labeling was performed with Akoya's Fluorescent Multiplex reagents according to the PerkinElmer´s application note 'Automated multiplex Biomarker Staining and Imaging' for CHIT1 (#HPA010575, Sigma, 1:1000), Iba1 (#019-19741, Wako, 1:10,000), PLP (#MCA839G, Bio-Rad, 1:5000), GFAP (#MS-1376, Thermo Scientific, 1:1000), TMEM119 (#HPA051870, Sigma, 1:1000) and CD68 (#M0814, Dakocytomation, 1:1000). In short, after deparaffination, the sections were fixed with 4% paraformaldehyde (PFA, 20 min), rinsed with deionized water and steamed in Antigen Retrieval buffer (pH 9.0, Akoya, AR900250ML) in a household food steamer (Braun, 1 h). The sections were rinsed with TBS-Tween 2.0 (TBST, pH 7.5) followed by a blocking step with Opal Antibody Diluent/Block solution (Akoya, ARD1001EA, 10 min). Next, we incubated the slides with a first primary antibody (4 °C, overnight). Then, the slides were rinsed with TBST and incubated with Akoya's secondary antibodies (Opal Polymer horseradish peroxidase conjugated antibodies, ARH1001EA, 10 min at RT). The sections were rinsed with TBST and incubated with a fluorophore (Opal dye 480, 520, 570, 620, 690 or 780, 1:100, 10 min) in 1X Plus Amplification Diluent (Akoya, FP1497). Next, the slides were rinsed with TBST, fixed with 4% PFA (10 min) and steamed with Antigen Retrieval buffer (AR6, Akoya, AR600250ML, 30 min). From here, the above-mentioned steps were repeated for each primary antibody. Finally, we counterstained with 4',6-Diamidino-2-phenylindole (DAPI) and the slides were mounted with Fluoromount-G (SouthernBiotech). The slides were scanned with a Vectra Polaris scanner (PerkinElmer) at 20× magnification or on a laser scanning microscope equipped with lasers for 504, 488, 543 and 633 nm excitation (Leica SP5, Leica, Mannheim, Germany).

## Statistical analysis

For single-time-point analysis and mixed-effects models, all statistical tests were performed with R v4.0.1. Normal distribution of the data was assessed with Shapiro-Wilk tests. Biomarker concentrations and clinical variables were normally distributed after a logarithmic transformation with base 10 and an inverse rank normal transformation, respectively.

**Single-time-point analysis.** To evaluate the relationships between the five microglia/macrophage CSF biomarkers – CHIT1, CHI3L1, sTREM2, GPNMB and CCL18 – and CSF NfL concentrations, we calculated pairwise Pearson correlation coefficients and tested their significance with two-sided Pearson correlation tests. Next, we tested the association between these biomarker concentrations and several MS disease activity parameters: ARMSS and ARR before treatment (primary outcomes); MSSS and EDSS (secondary outcomes). These associations were assessed in a linear regression model which included age at diagnosis and sex as covariates as well as the rs150192398 genotype for CHIT1 as in our previous work[7]. Based on the results of the linear regression model, we focused on the correlations between CSF CHIT1 concentrations and future disability (ARMSS, MSSS and EDSS). The percentage of variance explained ($r^2$) with and without clinical covariates, the division between MS patients with high (ARMSS ≥ 5) and low (ARMSS < 5) disability accumulation as well as the ROC curve were determined as previously described[7,61]. We applied Bonferroni corrections for multiple testing as stated in the table legends.

**Multi-time-point analysis – mixed-effects models.** The correlation between CSF CHIT1 concentrations at diagnosis and disability progression (EDSS) over time was assessed using both mixed-effects

models and artificial intelligence algorithms (see *Machine learning algorithms*). Mixed-effects models were chosen due to their robustness and power to handle unbalanced data given the gradual loss of EDSS data over time within our MS cohort. Despite the unbalanced EDSS data, the mixed-effects models allowed us to examine the impact of CHIT1 on disability progression over time while also considering variability within and across MS patients and time as well as controlling for relevant variables[81]. Within our model development, we tested models with and without the following considerations: controlling for 1) interindividual variance in EDSS scores; 2) intraindividual variance in EDSS scores across time; and 3) interaction effects between CHIT1 and time from MS diagnosis to EDSS score. Mixed-effects models were constructed and assessed with packages lme4 v1.1-31, afex v1.2-1 (convergence checks) and ggplot v3.4.2. Included continuous predictors were scaled to assist with convergence and result interpretation. Model fits were assessed using likelihood-ratio tests.

**Multi-time-point analysis – machine learning algorithms.** In D'hondt et al.[19], we developed machine learning algorithms to model EDSS trajectories of MS patients. In short, this approach transformed real-world EDSS assessments for each patient to yearly average disability scores up to the 10th year after diagnosis. These 10 disability averages then formed the outcomes of a machine learning model based on random forests. For each of these 10 outcomes, a random forest was trained. In a regressor chain fashion[82], the disability predictions of previous years were added as extra input to the random forests of the next years. These trained models could then be used to predict the disability of an individual MS patient for up to 10 years after diagnosis.

In this work, we used this regressor chain model to analyze the prognostic importance of input variables, in particular the CSF biomarkers under study. For this, we used permutation importance scores[20], which reflect the effect on model performance if one of the input variables is randomly shuffled (simulating the removal of that variable). Model performance (mean absolute error and Pearson correlation) was measured using 5-fold cross-validation (to avoid overfitting) and the variable shuffling was repeated multiple times to counter variability. In order to control for statistical interactions between CSF biomarkers, we conducted a separate analysis where we made a correction on the permutation importance scores for possible CSF biomarker interactions. To this end, in addition to shuffling each biomarker separately, we also shuffled all possible pairs, triplets, quartets and quintets of biomarkers (simulating the removal of several biomarkers simultaneously). In each of these shuffles, the performance drop was calculated and divided by the size of the shuffled subset of biomarkers (1, 2, 3, 4 or 5) and by the number of times each biomarker appears in a subset of that size (1, 5, 15, 5, 1). For each biomarker, a final score was then obtained by summing all performance drops in which that biomarker was shuffled. The machine learning methodology was developed in Python 3.8.10.

**Single-cell transcriptomics.** With regard to the transcriptomics, all statistical tests were performed in R v4.2.2. DGE results were consistently assessed for statistical significance using a two-sided unpaired Wilcoxon rank-sum test with Bonferroni correction for multiple testing. Differential expression of modules across clusters was demonstrated using a Kruskal-Wallis test followed by a two-sided Dunn's test for pairwise multiple comparisons with Benjamini-Hochberg correction for multiple testing. For pathway analysis, p-values were calculated by a one-sided Fisher's exact test with Benjamini-Hochberg correction. For the DGE analyses across trajectories, statistical significance was evaluated by means of a two-sided Wald test with Benjamini-Hochberg correction for multiple testing. The exact method is also always stated in the legends. For histology, no statistical analyses were performed.

## Reporting summary

Further information on research design is available in the Nature Portfolio Reporting Summary linked to this article.

## Data availability

All data that support the main findings in this study are available in the manuscript or the supplementary materials. Raw biomarker measurements are available in Supplementary Data 1. Source data are provided with this paper. The previously published CNS scRNA-seq and snRNA-seq datasets were downloaded from the Sequence Read Archive [SRA, National Center for Biotechnology Information (NCBI)] under BioProject accession numbers PRJNA544731 (Schirmer et al.[24]), PRJNA749443 (Absinta et al.[25]), PRJNA743676 (Miedema et al.[14]) and PRJNA726991 (Trobisch et al.[64]). Our previously unpublished CSF scRNA-seq cohort has been uploaded to the SRA under BioProject accession number PRJNA996357. Source data are provided with this paper.

## Code availability

The code scripts to recreate the main figures and results are accessible in Zenodo under https://doi.org/10.5281/zenodo.11235175.

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

## Acknowledgements

This study was funded by the National Multiple Sclerosis Society (International Progressive MS Alliance, PA-2002-36277, A. Goris), the

Dutch MS Research Foundation (Monique Blom - de Wagt grant, 22-1146 MS, SM), MS-Liga Vlaanderen (A. Goris) and the Flemish government (through the AI Research Program, CV). J.Be and R.D. hold PhD Fellowships of the Research Foundation-Flanders (FWO-Vlaanderen, 11A0523N and 1S38023N). B.D. holds a research mandate of KU Leuven (BOF-FKO, Bijzonder Onderzoeksfonds – Fundamenteel Klinisch Onderzoeker). Some of the resources and services used in this work were provided by the VSC (Flemish Supercomputer Center), funded by the Research Foundation - Flanders (FWO) and the Flemish Government. Figure 4A was partially created with BioRender.com. This study made use of publicly available datasets and we wish to sincerely thank the authors for sharing their data. In particular, we would like to thank S.M. Kooistra for providing us with additional metadata on their samples. We acknowledge all developers who contributed to the excellent tools we were able to use in our analyses. Finally, the authors thank A. Goris for help with conceptualizing this study, L. Yshii, M. David and D. De Wit for fruitful discussions, K. Clysters and C. Thys for their contribution in sample collection, as well as all patients and their families for their willingness to participate in this study.

## Author contributions

Conceptualization: J.Be, S.S., R.D., C.V., S.M., B.D. Methodology: J.Be, S.S., R.D., L.V., JBa, S.M. Investigation: J.Be, S.S., R.D., L.V., N.D., J.Ba, S.M. Visualization: J.Be, S.S., R.D., L.V., J.Ba, S.M. Funding acquisition: C.V., S.M., B.D. Supervision: P.M., J.Ba, C.V., S.M., B.D. Writing – original draft: J.Be, S.S., R.D., S.M. Writing – review & editing: J.Be, S.S., R.D., L.V., N.D., P.M., J.Ba, C.V., S.M., B.D.

## Competing interests

B.D. has received consulting fees from Bristol Myers Squibb, Almirall, Roche, Janssen, Novartis and Sanofi. All other authors declare that they have no competing interests.
