## [Peer Review File · Nature Communications]

CHIT1 at diagnosis predicts faster disability progression and reflects early microglial activation in multiple sclerosisReviewers' comments:

Reviewer #1 (Remarks to the Author):

In the manuscript presented by Beliën et al., they describe their analysis of CHIT1 as a potential biomarker in the prediction of the progression of MS and they analyse multiple pre-existing single-cell/single-nucleus datasets combined with newly generated single-cell data from the CSF. This combined approach is relevant and interesting and provides insights in cell states/activation patterns that might be important for progression of MS. However, I do have some concerns regarding the presentation and interpretation of the data that should be addressed before the study can be published.

The title is not exactly correct. With the data presented in the manuscript, where the majority of the single-cell data is derived from end-stage disease samples, it is not really possible to conclude on early microglial activation.

1. Fig 2a: the presentation of these data in 3D makes it very hard to interpret. For the individual datapoints it is impossible to determine where they are on some of the axis. Please consider other visual representation.
2. The model describing EDSS trajectories in Fig2B appears to be in contrast with the conclusion that CHIT1 levels at the time of diagnosis could be used to predict individual disease trajectories and instruct patients. The large degree of overlap between CHIT1 concentrations in the 3 groups at year 0 precludes predictions on an individual level.
3. Part of the single cell data, the data derived from cells in the CSF of MS patients, are newly collected. However, those data themselves, as well as their integration with the other datasets are not carefully described in the paper, while this is necessary.
4. A basic description of the data is lacking: How many cells were retrieved? What were the other cells in the CSF? According to Esualova et al. 2020 approx 4% of the cells in the CSF are transcriptionally similar to microglia. How was that in the current dataset? And how were the cells distributed over the 11 donors?
5. The classification of the cells from the CSF as microglia also needs more attention and clarification: In Schafflick et al. 2020, these cells are described to express markers of microglia, as well as markers of other macrophages and they considered them to be most similar to BAMs. Data that was used to define
6. The integration of the different datasets has to be described more extensively and also depicted in figures (can be supplemental). Where do the cells from the different datasets land? Do each of the clusters contain cells from each of the datasets? We already know that microglia from the cortex and spinal cord are different, how is that for cells from the CSF? And how well did the integration work?
7. Also, how are certain QC features distributed over the UMAP? How is the % of mitochondrial reads over all cells? Typically we find that clusters of cells with high mitochondrial reads end up in the middle of our UMAPs, which in this case could be overlapping with the cluster of cells that are

mostly expressing CHIT1. I am not saying that this is the case here, but I am unable to evaluate this, given that the QC basics after the integration are not presented.

8. The lack of insight in distribution of samples/locations of the cells in the UMAP makes it hard to interpret the rest of the data. If some of the clusters that are a bit further away in the UMAP (the ones labeled in yellow for example) are derived from a specific dataset, that makes it hard to decide what the subsequent trajectory analysis means. For example: I could imagine that the cells from the CSF are actually the clusters in yellow. Then it does not make sense to do the trajectory analysis as it is currently presented. Because it is driven by location and how likely is it that a cell from the CNS parenchyma would transform into a cell from the CSF. Again, I do not know if this is the case, since the data is not presented in the current manuscript.

9. The neuropathological evaluation that is depicted is taken from 1 sample and 1 lesion only, while in the methods it is stated that 11 active and 6 inactive lesions were analyzed from 6 patients. This analysis of all patients and all samples have to be presented. It is not a problem to show images of representative samples, but at the least a quantification of all should be included.

Reviewer #2 (Remarks to the Author):

In this manuscript Belien and colleagues investigate the prognostic role of CSF biomarkers in a longitudinal cohort of people with multiple sclerosis. They measured CSF levels of 5 biomarkers related to microglia/macrophages activity: chitinase 1 (CHIT1), chitinase-3-like protein 1 (CHI3L1), soluble triggering receptor expressed on myeloid cells 2 (sTREM2), glycoprotein non-metastatic melanoma protein B (GPNMB) and C-C motif chemokine ligand 18 (CCL18). Among these, CHIT1 is identified as a CSF biomarker of faster disability progression. Next, they show that CHIT1 is predominantly expressed by a distinct microglia subset in chronic active lesions enriched for pathways involved in lipid clearance and metabolism. Evaluation of post-mortem MS brain tissue confirmed that the CHIT1 is expressed by lipid-laden phagocytes in actively demyelinating lesions.

This is an interesting article well-presented which adds to the literature on this topic. Identification of robust biomarkers of disability progression in MS is a clinical unmet need. The role of microglia in demyelinating disease is of increasing interest and could have clinical implication for biomarker discovery and development of new therapies.

Some specific comments are the following:

- In the longitudinal cohort is not clearly specified if patients are treatment naïve at time of CSF collection
- Table 1 reports the characteristics of the study population and presents 3 sub cohorts. It is not clear what the two subcohorts indicated as disability and relapse are. Are these a sub group for which those clinical info were available?
- The raw data of the measurements of the 5 biomarkers that were tested were not included in the manuscript (or I could not find them...)
- Figure 4 and corresponding results: analyses were performed by the authors on a CSF cohort of 11

patients by scRNA seq and analyzed together with data available in the literature from snRNA-seq in brain/spinal cord tissues. There are some concerns about pulling together/harmonizing data from CSF and CNS tissues.

-could be important to add in the manuscript some information on the function of CHIT1

-Comments on neuropathological evaluation and Figure 7:

1) Data are presented in a qualitative way (e.g. CHIT1+ cell were abundantly expressed in actively demyelinating lesions, whereas inactive lesion only rarely..."); a more quantitative evaluation for the different lesions should be provided.

2) Add the histology of the MS lesion in Figure 7 and indicate which part of the lesion was then analyzed by immunofluorescence.

3) It is reported that CHIT1 + cells were generally Iba1 negative. Iba1 would be expected to be positive on microglia and macrophages, including foamy macrophages phagocytosing myelin. What would CHIT1+ and Iba1 negative cells be? The whole manuscript is about CHIT1 expression in microglia, but in this last piece it seems to be expressed on another cell type.

4) Staining with an isotype control antibody would be important to add.

Reviewer #3 (Remarks to the Author):

The manuscript by Beliën et al. is dealing with myeloid-related biomarkers for long-term disease outcome and treatment stratification in multiple sclerosis.

The authors analyzed five proteins expressed by microglia and/or macrophages in cerebrospinal fluid collected by diagnostic lumbar puncture in a large, longitudinal cohort of MS patients. Corroborating previous observations, they show that chitotriosidase (CHIT1) protein levels at diagnosis predict disability progression. Using single cell RNA sequencing of cells in the CSF and integration with previously published scRNA-seq or snRNA-seq datasets of CNS tissue from MS patients, the authors conclude that *CHIT1* is expressed by a subpopulation of activated microglia. *CHIT1*+ microglia seem to represent a transitional microglia state involved in lipid clearance and metabolism. Finally, the authors use immunofluorescence to localize CHIT1+ cells in actively demyelinating lesions.

While there might be some limitations regarding novelty (due to the authors' previous studies), the work is of high interest and translational impact, as proven prognostic markers for disease activity in MS are currently lacking. However, based on the presented data some of the main conclusions are premature and there are several major points of concern. Therefore, important questions regarding the identity of CHIT1+ cells and the proposed use of CHIT1 as a biomarker remain unresolved. These issues should be addressed to strengthen the study.

Major concerns:

1) The authors performed scRNAseq of cells from CSF and integrated their data with previously published scRNA-seq as well as snRNA-seq datasets of CNS tissue from MS patients. This integrated dataset is used to make conclusions about the expression of *CHIT1* in distinct microglia/macrophage subsets/states and different CNS/lesion regions or types. However, the contributions of the different studies to the final integrated dataset are not presented. Strong batch effects between different studies, isolation techniques, and type of transcriptomic profiling are expected and might explain some of the differences in *CHIT1* expression. As presented now, it is impossible to evaluate if the distinct clusters shown really indicate different microglia states or subsets. Are some of the clusters specific to certain datasets? Do the authors detect microglia-like cells in the CSF as others have previously reported (reviewed in Munro et al., 2022, <https://doi.org/10.1126/sciimmunol.abk0391>)? Is *CHIT1* expression related to specific myeloid subsets or enriched in certain datasets vs others? All of this is unclear and important to resolve, especially since sequencing depth differs between scRNA-seq and snRNA-seq.

2) Related to the previous point: according to the pseudotime trajectory analysis, proinflammatory and pre-activated clusters (MG8 and MG9, respectively) represent a distinct lineage in comparison to the other MS-enriched cluster (MG4-MG7, which arise from MG1-3 along three other trajectories). Without clearly describing possible batch effects and study origin of the distinct myeloid cells in their transcriptomic approach, this finding is surprising. Shouldn't "pre-activated" and pro-inflammatory states fall along a common trajectory at earlier pseudotime compared to more activated MS-associated states? These states are not sufficiently described and validated (see also major point 4).

3) How do the authors explain elevated *CHIT1* protein levels in the CSF of MS patients with predicted disease worsening? There is a brief sentence in the discussion speculating about microglial secretion and leakage of *CHIT1* protein into the CSF but the proposed mechanism is neither resolved nor properly discussed. Which myeloid cells could the authors detect in the CSF by their transcriptomic approach? While fewer cells outside of the lesion seem to express *CHIT1* mRNA, their localization might enable more efficient shedding into the CSF. Moreover, as the authors correctly state, transcript levels are not always correlated with protein expression levels. Further validation is needed, especially as some macrophages are reported to express *CHIT1* protein (Sjöstedt et al., 2020, <https://doi.org/10.1126/science.aay5947>).

4) The validation of *CHIT1* expression in relation to microglia/macrophage states by immunofluorescence seems incomplete. It is necessary to clarify the identity of the IBA1-TMEM119- GFAP- phagocytic *CHIT1*⁺ cells found in the lesion core. Considering the above mentioned concerns, further validation of the different myeloid clusters described by transcriptomic analysis in combination with *CHIT1* detection would help to firm up the conclusions. If possible, RNAscope detection of *CHIT1* transcripts might help to resolve the putative discrepancy of transcript versus protein localization.

Minor concerns:

1) Why did the authors choose to specifically investigate the five selected proteins as CSF biomarkers? There should be some clarification to justify this selection. It is difficult to judge the superiority of a myeloid protein over others as a biomarker when only 5 proteins in total were analyzed in detail in the study.

All reviewers' comments have been copied here in blue and we provide point-by-point responses in black to address all concerns raised.

Reviewer #1 (Remarks to the Author):

In the manuscript presented by Beliën et al., they describe their analysis of CHIT1 as a potential biomarker in the prediction of the progression of MS and they analyse multiple pre-existing single-cell/single-nucleus datasets combined with newly generated single-cell data from the CSF. This combined approach is relevant and interesting and provides insights in cell states/activation patterns that might be important for progression of MS. However, I do have some concerns regarding the presentation and interpretation of the data that should be addressed before the study can be published.

The title is not exactly correct. With the data presented in the manuscript, where the majority of the single-cell data is derived from end-stage disease samples, it is not really possible to conclude on early microglial activation.

We do acknowledge this reviewer's concern, considering that the vast majority of *CHIT1*+ cells were detected in CNS samples of more advanced/progressive MS patients. However, we chose to state 'early microglial activation' in the title to emphasize that we were able to measure CHIT1 in CSF already early in MS (at diagnostic LP) and that its concentration held prognostic value for the further disease course. Furthermore, with 'early microglial activation' we also allude to the early-activated, transitional cell state marked by *CHIT1* expression in the trajectory analysis.

In the revised manuscript, **we have now performed additional neuropathological evaluation in rare tissue samples from active lesions of five early-stage MS patients** with a disease duration between 0.2 and 5 months (Table S8). As discussed in the updated results section (from line 293 onwards) and indicated in the updated Table S8, we detected CHIT1+ cells in all active lesions from these early-stage MS patients, further strengthening our claim of CHIT1 being an early marker for microglial activation in MS.

1. Fig 2a: the presentation of these data in 3D makes it very hard to interpret. For the individual datapoints it is impossible to determine where they are on some of the axis. Please consider other visual representation.

We do concede that the 3D plot (Fig. 2A) is not the easiest to interpret. However, for that reason we also provided a summary graph (Fig. 2B) which shows the same information but plotted into a 2D space. For the sake of transparency, we decided to also retain the 3D graph, as our model relied on three variables. Moreover, we ensured that the easiest-to-interpret variables of the 3D plot (x and y; 'CHIT1 at diagnosis' and 'disability score') are complementary to the 2D plot where we displayed 'years since diagnosis' on the x-axis. In addition, a regression plane of the individual datapoints was provided to illustrate the global relationship between the three variables. We therefore feel that the complementary use of Fig. 2A and 2B is the best way to visualize these data.

2. The model describing EDSS trajectories in Fig2B appears to be in contrast with the conclusion that CHIT1 levels at the time of diagnosis could be used to predict individual disease trajectories and instruct patients. The large degree of overlap between CHIT1 concentrations in the 3 groups at year 0 precludes predictions on an individual level.

We would like to clarify that it is not possible to discern an overlap between CHIT1 concentrations in Fig. 2B. The overlap between the confidence intervals of the three curves reflects the overlap between EDSS values (the variable on the y-axis) at year 0. This is expected, since most MS patients will have a rather mild EDSS at diagnosis. The plot shows that MS patients with higher CSF CHIT1

concentrations at diagnostic LP are predicted to have faster disability progression as illustrated by the steeper slope of the green curve. The overlap between EDSS values at year 0 does not preclude the use of CSF CHIT1 concentration as a predictor of individual disability trajectories over the coming years. However, this study was indeed not designed to find potential clinical threshold values based on CSF CHIT1 concentrations. Like any biomarker, it is indispensable to validate our results in a large, independent, prospective cohort to consolidate its clinical potential. To meet this comment, **we rephrased the latter more explicitly in our discussion (line 454-457) and removed the premature reference to prediction at the individual level.**

3. Part of the single cell data, the data derived from cells in the CSF of MS patients, are newly collected. However, those data themselves, as well as their integration with the other datasets are not carefully described in the paper, while this is necessary.

We described the collection of our in-house CSF cohort in M&M (line 524-562). The clinical and sequencing-related characteristics of this cohort (and all publicly available datasets) are provided in Date file S2. This file also contains the number of cells (before and after QC and in the integrated Seurat object) per dataset as well as the median number of cells per patient, median number of genes per cell and median number of transcripts per cell – all listed per individual dataset. If any other specific details with regard to the dataset were omitted, we would gladly supply them. The method of integration of our CSF single-cell profiles with the other datasets is outlined in M&M (line 582-595). For transparency and reproducibility, our R code has already been uploaded to GitHub and will be made publicly available through Zenodo upon publication. Meanwhile, we have shared the R scripts with the editor at the moment of initial submission. In brief, we explored several integration strategies, including both Seurat v4 as well as Harmony with various (combinations of) covariates. We chose the strategy with the best balance between over- and underintegration, in line with the latest recommendations of the Single-cell Best Practices Consortium (Heumos *et al.* Nat. Rev. Gen. 2023). **We have now provided a more explicit expansion on this in M&M.**

Moreover, to cater to this comment and other comments from all three reviewers concerning the integration of datasets as well as to facilitate a comprehensive evaluation by future readers, **we have added a new supplementary figure (Fig. S2).** Here, we provide an overview of our integration of the myeloid cells derived from all five datasets. The UMAP plots visually show that our integration effectively removed technical variation (*i.e.* batch effects) such as the difference between single-cell profiles generated by either single-cell or single-nucleus sequencing, while retaining biologically relevant variation such as the different cellular composition of CSF and CNS datasets. The stacked bar plots illustrate both the absolute cell distribution and the relative cell distribution weighed to the total number of cells in each original dataset, tissue of origin, sequencing technique and MS subtype across all clusters. These bar plots show that there are no artificial clusters originating from a single dataset. We would like to emphasize that we added the UMAP and stacked bar plots split by disease course to be complete. However, their interpretation is not straightforward due to an uneven distribution of disease subtypes over the different datasets; e.g. all CSF samples were obtained from RRMS patients. **We have incorporated additional sentences in our results section referring to this figure (line 196-198).**

Finally, as cross-tissue integration remains challenging (Luecken *et al.* Nat. Methods 2022), one artefact resulting from this in our study might be our 'MMC' cluster where mDCs – primarily from the CSF – have been clustered together with microglia from the CNS datasets as we also acknowledge in the results section (line 203). Inversely, some CSF mDCs or other CSF myeloid cells with microglia-like features might – upon integration – have been pulled into some of the microglia clusters. We hypothesize that these cell types were clustered together due to increased expression of MHC class II-related molecules in a subset of microglia as described earlier (Masuda *et al.* Cell Rep. 2020), which therefore resembled the expression pattern of mDCs. Nevertheless, as readers can now evaluate in Fig. S2A, the overall composition of the CSF myeloid compartment after integration is in line with

expectations, with clear contributions to the monocyte and macrophage clusters, as well as to the MMC cluster, holding the mDCs. This composition is also in line with the work of Ostkamp *et al.* *Sci. Trans. Med.* 2022 who also combinedly analyzed myeloid cells from both CSF and CNS tissue and obtained a similar distribution of myeloid cells in the CSF compartment after integration (Fig. 2C in the Ostkamp paper). However, as our downstream focus in this manuscript is on characterizing *CHIT1+* cells and not on providing a detailed overview of the myeloid compartment in CSF and CNS tissue, we hope that by providing these additional supplementary analyses and figures, we could satisfy this and the other reviewer's concerns regarding our data integration.

4. A basic description of the data is lacking: How many cells were retrieved? What were the other cells in the CSF? According to Esaulova *et al.* 2020 app 4% of the cells in the CSF are transcriptionally similar to microglia. How was that in the current dataset? And how were the cells distributed over the 11 donors?

The number of cells retrieved was/is stated in the results section (line 193) and indicated per dataset in Data file S2 (before and after QC and in the integrated Seurat object). To isolate myeloid cells within each individual dataset, we performed preliminary clustering followed by manual (Data file S3) and automated annotation as described in M&M. Automated, unsupervised annotation with SingleR using the Novershtern reference [microarray datasets for sorted hematopoietic cell populations from GSE24759 (Novershtern *et al.* *Cell* 2011)] demonstrated that our CSF single-cell landscape in MS is in line with numerous previous reports on this topic (Ramesh *et al.* *PNAS* 2020, Esaulova *et al.* *Neurol-Neuroimmunol.* 2020, Schafflick *et al.* *Nat. Commun.* 2020, Straeten *et al.* *J. Neuroinfl.* 2022, Ostkamp *et al.* *Sci. Trans. Med.* 2022...). In the figures below, we show the automated annotation with SingleR (Novershtern reference) and the cell distribution per cluster across all 11 MS patients. We have provided these graphs in response to this reviewer's concerns, but we decided not to include them in the manuscript as, even though interesting, these particular results do not contribute anything new to the literature and other cell populations aside myeloid cells (e.g. B cells) are not relevant to the posed research questions.

Based on combined manual and automated annotation, we isolated the ‘pink’ and ‘dark green’ clusters (roughly annotated by SingleR as ‘monocytes’ and ‘myeloid dendritic cells’ respectively) for further analysis. Of note: as the Novershtern reference is a hematopoietic reference, CAMs and microglia-like cells present in the CSF were not annotated as such by this automated, unsupervised method, which is why manual annotation was added to ensure these cell types were present in the ‘pink’ and ‘dark green’ clusters before isolation. After integration of these CSF myeloid cells with those from the CNS datasets, a detailed myeloid subclustering and annotation was performed (Fig. 4B). As depicted in the new Fig. S2, CSF cells constituted a large, relative proportion of monocytes (MON) and myeloid dendritic cells (MMC). Certainly, some CSF myeloid cells clustered together with CNS-derived microglia, signifying that these cells at least portrayed a microglia-like phenotype/transcriptome. Where some studies such as Esaulova *et al.* *Neurol-Neuroimmunol.* 2020 and Ostkamp *et al.* *Sci. Trans. Med.* 2022 propose that at least a proportion of these CSF cells in MS patients can be classified as microglia, others prefer to refer to them as monocytes with a microglia character instructed by the CSF environment (Straeten *et al.* *J. Neuroinfl.* 2022) or to even use a more general term such as CSF macrophages (Ramesh *et al.* *PNAS* 2020). Our CSF macrophage landscape as annotated in Fig. S2 is in line with the composition described in a leading review in the field (Munro *et al.* *Sci. Immunol.* 2022), where CSF macrophages show subpopulations expressing typical genes for blood-derived monocytes, CAMs as well as CNS microglia. In our in-house CSF dataset, 5.3% (1192/22,506) of total CSF cells portrayed such a microglia-like phenotype.

However, we want to highlight that the focus on the CSF macrophages is perhaps less important in our study, since almost all *CHIT1*+ cells were found in CNS datasets (Fig. 4D and Fig. S4). This makes sense as we further showed their close association with active MS lesions (Fig. 4D and Fig. 7). The ability to measure *CHIT1* protein in the CSF seems to be less influenced by the presence of *CHIT1*-producing cells within the CSF and more – or almost exclusively – by activated CNS microglia in MS lesions. These microglia seem to secrete *CHIT1* into the CNS parenchyma, from where it might be subsequently washed out by the glia-lymphatic or ‘glymphatic’ system in which CSF mixes with the interstitial fluid to clear extracellular waste (Louveau *et al.* *Nature* 2015, Munro *et al.* *Sci. Immunol.* 2022), as has been described for β -amyloid (Iliff *et al.* *Sci. Trans. Med.* 2012). Of course, this mechanism is at this point still hypothetical and could form the topic of follow-up work.

5. The classification of the cells from the CSF as microglia also needs more attention and clarification: In Schafflick *et al.* 2020, these cells are described to express markers of microglia, as well as markers of other macrophages and they considered them to be most similar to BAMS. Data that was used to define

As mentioned in the prior comment and shown in Fig. S2, CSF cells constituted a large, relative proportion of monocytes (MON) and myeloid dendritic cells (MMC). The contribution of CSF cells to microglia (MG1-9) and CAM (CAM1 and 2) clusters was less dominant and more dispersed. However, the distinction between monocytes, (CNS-associated) macrophages and microglia solely based on transcriptomic data remains a challenge and there is no consensus yet in the field on unique canonical marker genes for each population (Paolicelli *et al.* Neuron 2022). Such lists are notoriously difficult to assemble, since microglia and macrophages are heterogeneous, plastic cells and scRNA-seq only captures a momentary transcriptional state of cells, with no insight into ontogeny or functionality as aptly stated by Ostkamp *et al.* Sci. Trans. Med. 2022.

Nonetheless, to confirm the robustness of our annotation we **added supplementary microglia-, CAM- and monocyte-associated module scores** (gene sets) and scored the enrichment for each individual cluster (Fig. S3 and Data file S3). The selected gene sets were derived from recent, high-impact research such as Van Hove *et al.* Nat. Neurosci. 2019, Jordão *et al.* Science 2019 and Sankowski *et al.* Nat Med. 2023. **We referred to these results in lines 203-208, 213-216 and 268-270.** We demonstrate that our initial separation into microglia, CAM and monocyte populations still holds. A large group of microglia clusters (in particular MG4-MG7) were annotated as 'disease-associated microglia' (DAM), which makes sense since all CSF and CNS samples were obtained from MS patients. In addition, we observed a gradual increase in the intensity of the DAM module scores from cluster MG2 up to cluster MG7, with cluster MG4 positioned in between which was already indicated by our trajectory analysis (Fig. 6).

6. The integration of the different datasets has to be described more extensively and also depicted in figures (can be supplemental). Where do the cells from the different datasets land? Do each of the clusters contain cells from each of the datasets? We already know that microglia from the cortex and spinal cord are different, how is that for cells from the CSF? And how well did the integration work?

We have detailed our integration strategy and answered the integration-related questions in the response to comment #3 (see above). With regard to the different types of tissue (CSF, brain and spinal cord), our objective was not to find any transcriptional differences between microglia in the CSF, brain and spinal cord. Since our goal was to find and phenotype *CHIT1*+ cells, regardless of tissue of origin, we integrated at the level of individual samples. Therefore, transcriptionally similar microglia derived from CSF, brain and spinal cord are clustered together in a single cluster and did not form separate clusters. However, some tissue types were more dominant in one cluster than another [e.g. CSF microglia in MG7 (Fig. S2)].

7. Also, how are certain QC features distributed over the UMAP? How is the % of mitochondrial reads over all cells? Typically we find that clusters of cells with high mitochondrial reads end up in the middle of our UMAPs, which in this case could be overlapping with the cluster of cells that are mostly expressing *CHIT1*. I am not saying that this is the case here, but I am unable to evaluate this, given that the QC basics after the integration are not presented.

All QC parameters that underwent correction as well as the used threshold values were described in M&M and are in line with the latest recommendations of the Single-cell Best Practices Consortium (Heumos *et al.* Nat. Rev. Gen. 2023). To enhance clarity and convenience for the reader, **we have now included these details in the results section as well** (line 190-193). All cells with a high percentage of mitochondrial RNA (mtRNA) content were removed based on published threshold values: a maximal percentage of mtRNA content of 15% for scRNA-seq data (Osorio *et al.* Bioinformatics 2021) and 5% for snRNA-seq data (Absinta *et al.* Nature 2021, Trobisch *et al.* Acta Neuropath. 2022). Moreover, after removal of cells that failed our QC criteria, we also regressed out the percentage mtRNA content with SCTransform as outlined in M&M (from line 569 onward), effectively limiting the effect it can have on downstream clustering.

However, in an effort to address any lingering concerns from this reviewer, we present the percentage of mtRNA content in a feature plot. We have not included this feature plot in the manuscript, because it is not common practice in scRNA-seq papers and the correction was already detailed in M&M. We are open to adding the illustration if the reviewer and/or editor deems it valuable.

In addition, an overview of basic QC parameters per dataset as well as the number of cells retained after QC were provided in Data file S2. **Now, we have more explicitly referred to Data file S2** in lines 189, 194, 528, 536, 553 and 579.

8. The lack of insight in distribution of samples/locations of the cells in the UMAP makes it hard to interpret the rest of the data. If some of the clusters that are a bit further away in the UMAP (the ones labeled in yellow for example) are derived from a specific dataset, that makes it hard to decide what the subsequent trajectory analysis means. For example: I could imagine that the cells from the CSF are actually the clusters in yellow. Then it does not make sense to do the trajectory analysis as it is currently presented. Because it is driven by location and how likely is it that a cell from the CNS parenchyma would transform into a cell from the CSF. Again, I do not know if this is the case, since the data is not presented in the current manuscript.

We have detailed our integration strategy and answered the integration-related questions in the response to comment #3 (see above). Moreover, we want to point out that our trajectory analysis was performed only with microglia clusters (MG1-9) as stated in the results section (line 271) and in M&M (from line 609 onward), specifically to avoid such nonsensical trajectories. CSF-predominant clusters such as monocytes (MON) and MMC were excluded as well as CAMs (CAM1 and 2) as the objective of the trajectory analysis was to determine the position of *CHIT1*+ microglia over pseudotime across the different microglial states.

9. The neuropathological evaluation that is depicted is taken from 1 sample and 1 lesion only, while in the methods it is stated that 11 active and 6 inactive lesions were analyzed from 6 patients. This analysis of all patients and all samples have to be presented. It is not a problem to show images of representative samples, but at the least a quantification of all should be included.

To cater to this comment, we **1) increased the number of MS patients and lesions stained** and **2) updated Table S8, now showing the percentages of CHIT1+ active/inactive lesions in the various MS cases we examined**. We also added a new supplementary figure (Fig. S5) showing CHIT1-immunoreactivity in different types of non-lesional and lesional tissue.

Reviewer #2 (Remarks to the Author):

In this manuscript Belien and colleagues investigate the prognostic role of CSF biomarkers in a longitudinal cohort of people with multiple sclerosis. They measured CSF levels of 5 biomarkers related to microglia/macrophages activity: chitinase 1 (CHIT1), chitinase-3-like protein 1 (CHI3L1), soluble triggering receptor expressed on myeloid cells 2 (sTREM2), glycoprotein non-metastatic melanoma protein B (GPNMB) and C-C motif chemokine ligand 18 (CCL18). Among these, CHIT1 is identified as a CSF biomarker of faster disability progression. Next, they show that CHIT1 is predominantly expressed by a distinct microglia subset in chronic active lesions enriched for pathways involved in lipid clearance and metabolism. Evaluation of post-mortem MS brain tissue confirmed that the CHIT1 is expressed by lipid-laden phagocytes in actively demyelinating lesions.

This is an interesting article well-presented which adds to the literature on this topic. Identification of robust biomarkers of disability progression in MS is a clinical unmet need. The role of microglia in demyelinating disease is of increasing interest and could have clinical implication for biomarker discovery and development of new therapies.

Some specific comments are the following:

- In the longitudinal cohort is not clearly specified if patients are treatment naïve at time of CSF collection

All MS patients were treatment naïve at time of CSF collection. CSF samples were obtained during diagnostic lumbar puncture, before formal diagnosis of MS and therefore patients were treatment naïve. **We added this information in the results section (line 82) as well as in M&M (line 504).**

-Table 1 reports the characteristics of the study population and presents 3 sub cohorts. It is not clear what the two subcohorts indicated as disability and relapse are. Are these a sub group for which those clinical info were available?

Exactly. **We clarified this in lines 84-85 of the results section.**

- The raw data of the measurements of the 5 biomarkers that were tested were not included in the manuscript (or I could not find them...)

The raw data of the measurements have been added as Data file S1 and is referred to in line 84 of the results section.

- Figure 4 and corresponding results: analyses were performed by the authors on a CSF cohort of 11 patients by scRNA seq and analyzed together with data available in the literature from snRNA-seq in brain/spinal cord tissues. There are some concerns about pulling together/harmonizing data from CSF and CNS tissues.

We have detailed our integration strategy and answered the integration-related questions in the response to comment #3 and #6 of Reviewer #1 (see above) – including concerns about cross-tissue integration.

-could be important to add in the manuscript some information on the function of CHIT1

We have elaborated on this in the discussion (line 386-399).

-Comments on neuropathological evaluation and Figure 7:

1) Data are presented in a qualitative way (e.g. CHIT1+ cell were abundantly expressed in actively demyelinating lesions, whereas inactive lesion only rarely..."); a more quantitative evaluation for the different lesions should be provided.

Kindly refer to our reply to comment #9 of Reviewer #1.

2) Add the histology of the MS lesion in Figure 7 and indicate which part of the lesion was then analyzed by immunofluorescence.

We added this information to Fig. 7. as panel A. We also added a new supplementary figure (Fig. S5) showing CHIT1-immunoreactivity in different types of non-lesional and lesional tissue.

3) It is reported that CHIT1 + cells were generally Iba1 negative. Iba1 would be expected to be positive on microglia and macrophages, including foamy macrophages phagocytosing myelin. What would CHIT1+ and Iba1 negative cells be? The whole manuscript is about CHIT1 expression in microglia, but in this last piece it seems to be expressed on another cell type.

Apart from our detailed [and now also extended (see comment #5 of Reviewer #1)] annotation of CHIT1+ cells as microglia on the transcript level, the best proof of their microglial identity in the histology part is the fact that the cells that have little CHIT1, still are TMEM119+ (Fig. 7 and described in the results section). TMEM119 is one of the main homeostatic microglia markers that is not expressed on macrophages from the periphery and has been shown to get downregulated in active MS lesions and other inflammatory conditions of the brain (Zrzavy *et al.* Brain 2017, Zrzavy *et al.* Brain Pathol. 2018, van Wageningen *et al.* Acta Neuropathol. Commun. 2019). Moreover, decreased Iba1 expression and even Iba1-negative microglia have also previously been described in neurodegenerative diseases such as Huntington's and Alzheimer's disease (Lier *et al.* Cells 2021). Keren-Shaul *et al.* Cell 2017 showed that DAMs found in a mouse model of Alzheimer's disease were characterized by a downregulation of AIF1 (the gene encoding for Iba1) as well as homeostatic genes such as TMEM119 and P2RY12.

Moreover, **we have now also added a CD68 stain**, showing that whilst the CHIT1+ cells seem to lose Iba1 and TMEM119 expression upon continuous PLP accumulation, they remain CD68+. This combined with the total absence of immunoreactivity for GFAP, strengthens the claim that these CHIT1+ cells are indeed phagocytic CNS microglia/macrophages and not another cell type such as astrocytes.

We have now elaborated on all of this more clearly in the results section (from line 293 onwards) **and in the discussion** (line 406-418) in the revised manuscript.

4) Staining with an isotype control antibody would be important to add.

As stated in M&M, the anti-CHIT1 antibody is a polyclonal rabbit antibody rather than a mouse isotype-specific antibody and thus it is not possible to provide an isotype-specific control. Omission of the CHIT1 antibody however shows an absence of staining. Further, in the multiplex fluorescence stainings, CHIT1 was used in combination with several other primary rabbit antibodies such as Iba1, GFAP and TMEM119. As can be appreciated in Fig. 7, all of these stainings have their own target and staining patterns do not overlap, thus guaranteeing that in this multiplex staining the specificity of these antibodies is secured.

Reviewer #3 (Remarks to the Author):

The manuscript by Beliën *et al.* is dealing with myeloid-related biomarkers for long-term disease outcome and treatment stratification in multiple sclerosis.

The authors analyzed five proteins expressed by microglia and/or macrophages in cerebrospinal fluid collected by diagnostic lumbar puncture in a large, longitudinal cohort of MS patients. Corroborating previous observations, they show that chitotriosidase (CHIT1) protein levels at diagnosis predict disability progression. Using single cell RNA sequencing of cells in the CSF and integration with previously published scRNA-seq or snRNA-seq datasets of CNS tissue from MS patients, the authors conclude that *CHIT1* is expressed by a subpopulation of activated microglia. *CHIT1*+ microglia seem to represent a transitional microglia state involved in lipid clearance and metabolism. Finally, the authors use immunofluorescence to localize CHIT1+ cells in actively demyelinating lesions.

While there might be some limitations regarding novelty (due to the authors' previous studies), the work is of high interest and translational impact, as proven prognostic markers for disease activity in MS are currently lacking. However, based on the presented data some of the main conclusions are premature and there are several major points of concern. Therefore, important questions regarding the identity of CHIT1+ cells and the proposed use of CHIT1 as a biomarker remain unresolved. These issues should be addressed to strengthen the study.

With regard to the concern about the novelty of this work in comparison to our previous study, we would like to clarify that we did not only corroborate our previous observations in the first part of this manuscript. Our previous paper (Oldoni *et al.* *Annal. Neurol.* 2020) provided a first hint at the potential of CHIT1 as a prognostic marker by analyzing a smaller cohort and a single-time-point disability assessment at a median of 5 years after diagnosis. In the current work, we performed longitudinal analyses in a larger cohort with multi-time-point disability assessments. We used 'classical' biostatistics in conjunction with a novel machine learning model, allowing for the first time the identification of CHIT1 as a marker, not only for projected outcome but also for the pace of disability progression itself. With regard to MS clinical practice, the added value of multi-time-point over single-time-point disability assessments is best illustrated with an example: It is possible that an MS patient has only one severe relapse with poor recovery. Although this MS patient might have no (or very low) disease activity in the next few years and is therefore stable, the disability outcome measured 5 years later is high. On the other hand, multi-time-point disability assessments enabled us to measure disability progression over time, i.e. the MS patient experiences increments in disability over time and is not stable. The key point is that our previous single-time-point disability assessment could not discriminate between the stable MS patient with a high EDSS after one severe relapse with poor recovery or an MS patient with disability progression over time; multi-time-point disability assessment can make this distinction. There is a vast heterogeneity in the pace of disability progression and CHIT1 measurements might help clinicians in the stratification of DMTs as well as inclusion in novel clinical trials already at the time of diagnosis. Moreover, **in the current manuscript we provided the first biological insights into where this prognostic potential of CHIT1 might come from using both newly-generated scRNA-seq data as well as publicly available data and neuropathological validation. We now also expanded on the latter in this revised manuscript including rare tissue samples of lesions from early-stage MS patients (disease duration of 0.2-5 months).**

Major concerns:

1) The authors performed scRNAseq of cells from CSF and integrated their data with previously published scRNA-seq as well as snRNA-seq datasets of CNS tissue from MS patients. This integrated dataset is used to make conclusions about the expression of CHIT1 in distinct microglia/macrophage subsets/states and different CNS/lesion regions or types. However, the contributions of the different studies to the final integrated dataset are not presented. Strong batch effects between different studies, isolation techniques, and type of transcriptomic profiling are expected and might explain some of the differences in *CHIT1* expression. As presented now, it is impossible to evaluate if the distinct clusters shown really indicate different microglia states or subsets. Are some of the clusters

specific to certain datasets? Do the authors detect microglia-like cells in the CSF as others have previously reported (reviewed in Munro et al., 2022, <https://doi.org/10.1126/sciimmunol.abk0391>)? Is *CHIT1* expression related to specific myeloid subsets or enriched in certain datasets vs others? All of this is unclear and important to resolve, especially since sequencing depth differs between scRNA-seq and snRNA-seq.

We have detailed our integration strategy and answered the integration-related questions in the response to comment #3 and #6 of Reviewer #1 (see above). Supplementary analyses showed that our integration strategy effectively removed technical variation (*i.e.* batch effects) such as the difference between single-cell profiles generated by either single-cell or single-nucleus sequencing, while retaining biologically relevant variation such as the different cellular composition of CSF and CNS datasets. **Our integration strategy also prevented the formation of any artefactual dataset-specific clusters (Fig. S2).**

Moreover, with regard to the final part of this comment, **we also now included Fig. S4 showing the distribution of *CHIT1*+ cells over the individual datasets.** These analyses clarified that our *CHIT1* signal is not originating from a single dataset. All datasets seem to contain at least some *CHIT1*+ cells, with the relative contribution of each dataset determined by biological characteristics of the tissue/lesion type analyzed as illustrated in Fig. 4D and S2. We have referred to this new supplementary figure in the results section (line 215).

2) Related to the previous point: according to the pseudotime trajectory analysis, proinflammatory and pre-activated clusters (MG8 and MG9, respectively) represent a distinct lineage in comparison to the other MS-enriched cluster (MG4-MG7, which arise from MG1-3 along three other trajectories). Without clearly describing possible batch effects and study origin of the distinct myeloid cells in their transcriptomic approach, this finding is surprising. Shouldn't "pre-activated" and pro-inflammatory states fall along a common trajectory at earlier pseudotime compared to more activated MS-associated states? These states are not sufficiently described and validated (see also major point 4).

In order to check the robustness of the annotation of our myeloid subclustering, we performed supplementary analyses and provided a new supplementary Fig. S3 as outlined in comment #5 of Reviewer #1, as well as further histological characterization. And while it would be interesting to even more deeply characterize all our different myeloid/microglia clusters both on the transcriptomic and proteomic level, we find it reasonable to assert that such an endeavor goes beyond the scope of this study. With the advent of high-dimensional -omics methods, the field of myeloid/microglial heterogeneity has expanded significantly and various terms for different cell states have been coined. Recently, efforts have been undertaken to harmonize this research field and nomenclature (Masuda *et al.* Cell Rep. 2020, Paolicelli *et al.* Neuron 2022). Here, we attempted to stay as close as possible to these latest reference works, with a primary focus on the microglial cell state of interest - the one producing our putative biomarker. After all, our objective for the single-cell transcriptomics part was to provide a biological rationale for *CHIT1* as a prognostic biomarker in MS clinical practice. In line with such developments and recommendations, we did not want to introduce yet another set of annotations for our clusters and scored module (gene set) enrichment per cluster in an unbiased way, adopting the terminology used in the original papers as the names of our modules. Based on this rationale, we grouped the names of our modules to describe the three main cluster clouds in Fig. 6B, as also indicated in the figure legend.

To solidify more objectively what was already visually discernable, *i.e.* that our *CHIT1*+ cells seem to be positioned between the more homeostatic and the more damage-/MS-associated microglia, we turned to pseudotime trajectory analysis using slingshot. As output, slingshot provides both lineages and curves. Slingshot's first stage uses a cluster-based MST (minimum spanning tree) to stably identify the key elements of the global lineage structure, *i.e.* the number of lineages and where they branch. For the second stage, a method called 'simultaneous principal curves' is used to fit smooth

branching curves to these lineages, thereby translating the knowledge of global lineage structure into stable estimates of the underlying cell-level pseudotime variable for each lineage (Street *et al.* BMC Genomics 2018).

In the manuscript, we included the UMAP plot showing the curves. Here, we added the UMAP plot showing the lineages for the sake of discussion. The individual trajectories are less discernable here, which is why we chose to include the curve visualization in the manuscript. However, this graph stresses the fact that *CHIT1*+ MG4 forms an intermediate cluster, after which the trajectories seem to split up towards the damage-/MS-associated microglia (MG5 and 6) on the one hand and the pre-activated/pro-inflammatory clusters (MG8 and 9) on the other hand as rightfully mentioned by this reviewer. What this trajectory pattern, together with our other results, suggests is that homeostatic microglia upregulate *CHIT1* when they start to get activated and involved in myelin clearance. However, its expression seems to wane again when microglia start to develop a damage-/MS-associated phenotype, perhaps instigated by unremitting inflammatory cues and/or continuous lipid phagocytosis and accumulation. The second trajectory might reflect microglia present in the MS brain that obtain a more general activated/inflammatory profile, perhaps as a form of bystander activation – possibly explaining our dichotomous trajectory pattern.

3) How do the authors explain elevated *CHIT1* protein levels in the CSF of MS patients with predicted disease worsening? There is a brief sentence in the discussion speculating about microglial secretion and leakage of *CHIT1* protein into the CSF but the proposed mechanism is neither resolved nor properly discussed. Which myeloid cells could the authors detect in the CSF by their transcriptomic approach? While fewer cells outside of the lesion seem to express *CHIT1* mRNA, their localization might enable more efficient shedding into the CSF. Moreover, as the authors correctly state, transcript levels are not always correlated with protein expression levels. Further validation is needed, especially as some macrophages are reported to express *CHIT1* protein (Sjöstedt *et al.*, 2020, <https://doi.org/10.1126/science.aay5947>).

The different subquestions in this comment have been answered with the additional analyses and data provided:

- Which (myeloid) cells do we detect in the CSF? → Fig. S2. Also see our response to comment #4 of Reviewer #1.
- Moreover, **we have now elaborated on the putative function of CHIT1 in the discussion (line 386-399) as well as on our rationale behind the elevated CSF CHIT1 concentrations in MS patients at diagnostic lumbar puncture (line 426-433 and in our response to comment #4 of Reviewer #1).**

4) The validation of CHIT1 expression in relation to microglia/macrophage states by immunofluorescence seems incomplete. It is necessary to clarify the identity of the IBA1- TMEM119- GFAP- phagocytic CHIT1+ cells found in the lesion core. Considering the above mentioned concerns, further validation of the different myeloid clusters described by transcriptomic analysis in combination with CHIT1 detection would help to firm up the conclusions. If possible, RNAscope detection of *CHIT1* transcripts might help to resolve the putative discrepancy of transcript versus protein localization.

In the revised manuscript, we have now significantly expanded our neuropathological validation. **Not only did we include rare tissue samples of five early-stage MS patients, we also provided a more quantitative overview of the CHIT1 immunoreactivity of the different lesions (Table S8) and we included a CD68 stain as well as a new supplementary figure (Fig. S5) showing CHIT1-immunoreactivity in different types of non-lesional and lesional tissue.** We explained our efforts to further validate and consolidate the phenotype of CHIT1+ microglia in comment #3 of the neuropathology comments of Reviewer #2.

Furthermore, **our extended histological analyses** also allowed us to shed some light on the putative discrepancy of transcript *versus* protein localization that was initially present in the manuscript. For this we refer to the **updated results section of the histology** (from line 293 onwards) **as well as the discussion** (line 406-418 and line 449-452).

Minor concerns:

1) Why did the authors choose to specifically investigate the five selected proteins as CSF biomarkers? There should be some clarification to justify this selection. It is difficult to judge the superiority of a myeloid protein over others as a biomarker when only 5 proteins in total were analyzed in detail in the study.

Our selection of myeloid markers is justified by the rising interest in and knowledge of CNS macrophages in MS pathogenesis as well as the rising interest and promise of microglial biomarkers for MS disease activity as stated in the introduction (Kent *et al.* Nat. Rev. Immunol. 2024, Distéfano-Gagné *et al.* Nat. Rev. Neurosci. 2023, Guerrero *et al.* Front. Immunol. 2020). The choice of these five specific markers is based on previous work from our group (Oldoni *et al.* Annal. Neurol. 2020) as well as preliminary literature on their diagnostic or prognostic potential in MS or other neurological disorders, **as now also stated in the manuscript** (lines 75-78).

REVIEWER COMMENTS

Reviewer #1 (Remarks to the Author):

Dear editors at Nature Communications,

I appreciate the effort of the authors to clarify their work. Given the inclusion of several data elements describing the integration of multiple sn/scRNAseq datasets, the interpretation of the data is now easier and available to all readers.

I can now recommend publication of the manuscript

a minor comment: the data file s2 was missing from the downloadportal. I am assuming they were not altered from the previous version of the manuscript

Reviewer #2 (Remarks to the Author):

The authors have addressed my comments in a satisfactory way. In my opinion the article can be published in the current revised form.

Reviewer #3 (Remarks to the Author):

In their revised manuscript the authors addressed some major concerns by including a better description of the single-cell/-nucleus transcriptomic part and additional neuropathological data. However, several points have not been sufficiently resolved to recommend dissemination to a broad audience.

I can accept the authors' response regarding novelty related to the putative implications for clinical practice. However, the "first biological insights" into where the prognostic potential of CHIT1 might come from are not fully convincing at present.

1) The authors have now presented their transcriptomic approach in much better detail. However, this shows that CHIT1 is almost absent from their CSF single-cell dataset and undermines the necessity of this dataset for novel insights on the origin of CHIT1. It might have been similarly helpful to analyze/integrate previous data on CSF cells from MS patients at single-cell level (e.g., Ramesh et al. PNAS 2020, Esaulova et al. Neurol-Neuroimmunol. 2020, Schafflick et al. Nat. Commun. 2020, Straeten et al. J. Neuroinfl. 2022, Ostkamp et al. Sci. Trans. Med. 2022).

2) I appreciate that the authors refrain from introducing even more microglia acronyms and understand that not all the observed microglia clusters can be carefully validated and characterized in the present work.

However, it is necessary to better validate at least some of the proposed early and late reactive microglia states to substantiate their conclusion that CHIT1 is expressed on a distinctive transitional state related to lipid degradation and metabolism (see below).

It would also be helpful to include the putative explanations the authors offered in the rebuttal regarding the pseudotime trajectory analysis into the Discussion and include the presented slingshot lineages in the Supplementary Data.

3) Resolved

4) In the revised manuscript it is now clear that CSF myeloid cells (mostly monocytes and dendritic cells, some microglia-like cells) do not substantially contribute to CHIT1 levels. Again, this makes a conclusive characterization of the CHIT1+ microglia in smoldering/active lesions one of the most important aspects of this study.

The authors present data suggesting that CHIT1 is present on IBA1+ cells with limited PLP uptake that appear smaller and supposedly express the transmembrane protein TMEM119. However, the TMEM119 immunofluorescence is not convincing since it often does not appear to be present on the cell surface but intracellularly. There are also many TMEM119 immunoreactive spots in the images that do not colocalize with IBA1+ cells (microglia/macrophages).

IBA1 is considered a general microglia/macrophage marker and is usually expressed on homeostatic microglia, reactive microglia, as well as macrophages. The authors find many larger CHIT1+ cells that do not colocalize with IBA1 but with CD68 and argue that previous studies have observed a downregulation of IBA1 by some (late) reactive states of microglia. In addition, they describe that AIF1 is downregulated in the integrated sn/scRNA-seq dataset. However, the only cluster with downregulation of AIF1 (other clusters show upregulation) is MG9 which shows virtually no CHIT1 expression. Moreover, the CHIT1+IBA1-CD68+ cells are full of PLP and appear strongly phagocytic, arguing against the conclusion that CHIT1 is mostly derived from microglia at a transitional state.

A more detailed quantification of the frequencies of CHIT1 reactivity (IBA1+/-, TMEM119+/-, CD68+/-, PLP+/-) in different lesions is missing.

The used selection of markers does not help to pinpoint the CHIT1 reactivity to the MG4 cluster and the previously mentioned discrepancies between RNA and protein levels have not been resolved.

Finally, considering some of the discrepancies between the presented data and the conclusions, it is necessary to validate the specificity and reliability of the used antibodies.

All reviewers' comments have been copied here in blue and we provide point-by-point responses in black to address all concerns raised. Major adaptations to the paper are indicated in bold here and with track changes in the manuscript itself.

Reviewer #1 (Remarks to the Author):

Dear editors at Nature Communications,

I appreciate the effort of the authors to clarify their work. Given the inclusion of several data elements describing the integration of multiple sn/scRNAseq datasets, the interpretation of the data is now easier and available to all readers.

I can now recommend publication of the manuscript

a minor comment: the data file s2 was missing from the download portal. I am assuming they were not altered from the previous version of the manuscript

We would like to thank Reviewer #1 for their efforts and insightful comments and we appreciate their recommendation to publish our work.

Data file S2 (now Supplementary Data 2) was indeed not altered during the revisions.

Reviewer #2 (Remarks to the Author):

The authors have addressed my comments in a satisfactory way. In my opinion the article can be published in the current revised form.

We would like to thank Reviewer #2 for their efforts and insightful comments and we appreciate their recommendation to publish our work.

Reviewer #3 (Remarks to the Author):

In their revised manuscript the authors addressed some major concerns by including a better description of the single-cell/-nucleus transcriptomic part and additional neuropathological data. However, several points have not been sufficiently resolved to recommend dissemination to a broad audience.

I can accept the authors' response regarding novelty related to the putative implications for clinical practice. However, the "first biological insights" into where the prognostic potential of CHIT1 might come from are not fully convincing at present.

We thank Reviewer #3 for their thorough review of our revised manuscript and to prompt us to reflect even further on the more complex parts of our work, resulting now in a more solid narrative. In the following paragraphs we explain our rationale in great detail, hoping to satisfy any remaining doubts regarding the first biological insights provided in the final part of our manuscript.

1) The authors have now presented their transcriptomic approach in much better detail. However, this shows that CHIT1 is almost absent from their CSF single-cell dataset and undermines the necessity of this dataset for novel insights on the origin of CHIT1. It might have been similarly helpful to analyze/integrate previous data on CSF cells from MS patients at single-cell level (e.g., Ramesh et al. PNAS 2020, Esaulova et al. Neurol-Neuroimmunol. 2020, Schafflick et al. Nat. Commun. 2020, Straeten et al. J. Neuroinfl. 2022, Ostkamp et al. Sci. Trans. Med. 2022).

Indeed, from our transcriptomic approach we were able to conclude that the CNS is the main source of CHIT1-expressing cells as our CSF dataset contained very few CHIT1+ cells. However, as we were able to measure the CHIT1 protein (using ELISA) in the CSF of our biomarker cohort, it was an obvious

2) I appreciate that the authors refrain from introducing even more microglia acronyms and understand that not all the observed microglia clusters can be carefully validated and characterized in the present work.

However, it is necessary to better validate at least some of the proposed early and late reactive microglia states to substantiate their conclusion that *CHIT1* is expressed on a distinctive transitional state related to lipid degradation and metabolism (see below).

We elaborate on the transitional cell state of *CHIT1*⁺ microglia at the transcript and protein level in 4) below.

It would also be helpful to include the putative explanations the authors offered in the rebuttal regarding the pseudotime trajectory analysis into the Discussion and include the presented slingshot lineages in the Supplementary Data.

We accommodated this request by adding slingshot's global lineage structure, which we presented in the first round of revisions, as an **extra supplementary figure (Fig. S5B)**. Moreover, we provided a more detailed overview of the four individual trajectories/lineages in slingshot's smoothed curves representation of our pseudotime trajectory analysis (**Fig. S5A**). Here, the divergence of the curves after cluster MG4 towards the 'MS-associated' clusters (MG5 and 6) on the one hand (lineage 2 and 3) and the 'pre-activated'/'pro-inflammatory' clusters (MG8 and 9) on the other hand (lineage 1) are more clearly depicted and in accordance with the global lineage structure in Fig. S5B. This is now also stated in the results section (line 266-274).

We also included the **putative biological explanation** of the divergent trajectories into the **discussion** (line 415-426). However, this explanation remains speculative, as we state in the manuscript, and the most important take-away from our trajectory analysis is that *CHIT1* expression accompanies the transition from a homeostatic towards a more activated, MS-associated cell state in microglia – detailed in 4) below.

3) Resolved

We thank the reviewer for their appreciation of our efforts to resolve this comment.

4) In the revised manuscript it is now clear that CSF myeloid cells (mostly monocytes and dendritic cells, some microglia-like cells) do not substantially contribute to *CHIT1* levels. Again, this makes a conclusive characterization of the *CHIT1*⁺ microglia in smoldering/active lesions one of the most important aspects of this study.

The most important aspect of this study, as well as the biggest contribution to the field, resides in the fact that we are the first to identify *CHIT1* as a putative CSF biomarker for disability progression in a large cohort of MS patients with longitudinal disability follow-up, using both advanced statistical modelling, as well as a custom-made machine learning model. Of course, we do acknowledge the value of functional insights behind putative clinical tools, which is why we added the extensive transcriptomics and protein-level work in the second part of our paper. In the transcriptomics, we describe *CHIT1* to be primarily expressed by a distinct microglia subset that is most abundant within active MS lesions and is enriched for pathways related to foam cell differentiation, lipid homeostasis and clearance. Moreover, our trajectory analysis illustrates that *CHIT1* expression in microglia accompanies the transition from a homeostatic towards a more activated, MS-associated phenotype. In the neuropathological evaluation of both early- and late-stage MS patients, we then validate our most important transcriptomic findings on the protein level, namely that 1) *CHIT1*⁺ cells are indeed present in almost all actively demyelinating lesions, whereas completely absent in inactive lesions, 2) *CHIT1*⁺ cells are PLP-laden and 3) that with the acquirement of *CHIT1* immunoreactivity, microglia seem to lose characteristic homeostatic markers, indicating transition to an activated cell state.

We do acknowledge that there are still unanswered questions and that now more in-depth neuropathology as well as functional cellular studies are needed to primarily 1) further determine the

exact location of CHIT1 cells within different lesions types and 2) infer their exact phenotype and role in the de- and/or remyelination processes in MS. **However, while we are currently undertaking such efforts, we strongly believe that answering these questions falls outside the scope of this particular manuscript as it would constitute a new study in itself with its own research questions, shifting the focus away from the biomarker identification.** Furthermore, it would significantly delay the dissemination of our findings to the scientific community and would not add substantial evidence to support the main message of our paper: the potential of CHIT1 as an early prognostic CSF biomarker in MS.

To acknowledge this more outspokenly in the manuscript itself, we have added line 481-485 to the discussion.

The authors present data suggesting that CHIT1 is present on IBA1+ cells with limited PLP uptake that appear smaller and supposedly express the transmembrane protein TMEM119. However, the TMEM119 immunofluorescence is not convincing since it often does not appear to be present on the cell surface but intracellularly. There are also many TMEM119 immunoreactive spots in the images that do not colocalize with IBA1+ cells (microglia/macrophages).

There seems to be some doubt on the part of the reviewer with regard to the validity of our TMEM119 staining. To address these concerns, we would like to elaborate on this staining. First, we want to emphasize that CHIT1 is indeed clearly present on small Iba1+ TMEM119+ cells (now Fig. 7E), which shows that these cells are in fact microglia. We believe it is an understatement to describe these cells as “supposedly” expressing TMEM119, as the immunoreactivity in the staining is very clear (Fig. 7E). Furthermore, the antibody that was used (Sigma #HPA051870) has been used in many publications by us (i.e. Zrzavy *et al.*, Brain 2017 and Brain 2019) and by others (i.e. Krasemann *et al.*, Immunity 2017). Secondly, regarding the intracellular expression of TMEM119, we agree that TMEM119 within the lesions seems to be expressed intracellularly. This however does not negate the validity of our TMEM119 staining as, during activation, TMEM119 is abundantly cleaved and redistributed intracellularly to the phagosomes (Ruan & Elyaman, Front. Cell. Neurosci. 2022). Outside lesions, in NAWM, TMEM119 and Iba1 are co-expressed on microglial cells in a more resting state. Here, TMEM119 also appears to be expressed on the surface of thin processes. **To show the different location (surface versus intracellular) and intensity of TMEM119 inside and outside lesions to the reviewer and reader, we now included an image of TMEM119 with Iba1 and CHIT1 in resting microglia in the NAWM in Figure 7 (new Fig. 7A).**

With regard to the TMEM119 immunoreactivity that does not exactly colocalize with IBA1+ cells, we also agree that this seems to be the case in some cells. **Moreover, this non-overlapping also seems to be present in resting microglia as now can be seen in the new Fig. 7A.** Although we do not exactly know the reason for this, similar observations can be found in a myriad of other studies (Zrzavy *et al.*, Brain 2017, Fig. 1O&P or van Wageningen *et al.*, Acta Neuropath. Comm. 2019 Fig. 3F&I). A first possible explanation could be that Iba1 and TMEM119 may not be distributed equally on the surface. Secondly, we think that it might at least in part technically result from the strong amplification of signals by Akoya’s multiplex system staining with OPAL dyes. This results in strong differences in signal intensity, especially in microglia in lesions that have downregulated TMEM119.

IBA1 is considered a general microglia/macrophage marker and is usually expressed on homeostatic microglia, reactive microglia, as well as macrophages. The authors find many larger CHIT1+ cells that do not colocalize with IBA1 but with CD68 and argue that previous studies have observed a downregulation of IBA1 by some (late) reactive states of microglia. In addition, they describe that AIF1 is downregulated in the integrated sn/scRNA-seq dataset. However, the only cluster with downregulation of AIF1 (other clusters show upregulation) is MG9 which shows virtually no CHIT1 expression. Moreover, the CHIT1+IBA1-CD68+ cells are full of PLP and appear strongly phagocytic, arguing against the conclusion that CHIT1 is mostly derived from microglia at a transitional state.

We thank the reviewer for the in-depth and critical appraisal of our manuscript and would like to nuance the ‘transitional cell state’ of *CHIT1*⁺ microglia. Our trajectory analysis demonstrated that cluster MG4 is positioned around pseudotime value 7.5, located in-between homeostatic microglia clusters at the start of the lineages and more activated, MS-associated microglia clusters at the end of the lineages. We also showed that around that same pseudotime value 7.5 (and a bit further) *CHIT1* expression is increased, in particular in lineage 3 (Fig. 6). Indeed, we observed that *CHIT1* expression peaked in cluster MG4 and that most *CHIT1*⁺ cells resided within cluster MG4 (Fig. 4). **We have now added a new supplementary data file (Supplementary Data 6) with all the DEGs between the start and end points of the trajectories, as well as between the start and middle point (pseudotime value 7.5) of the trajectories.** One of these DEGs showing the same pattern as *AIF1* is homeostatic microglia gene *CX3CR1* which is significantly downregulated (after Benjamini-Hochberg correction) from start to middle point and from start to end point across trajectories (in top 3 of genes). *CX3CR1* has been described as one of the classic homeostatic genes in human microglia that is downregulated in microglia that exit the homeostatic cell state and enter a more activated cell state (Li *et al.*, Cell Stem Cell 2022). At the protein level, small *CHIT1*⁺ cells are TMEM119⁺ and Iba1⁺, whereas larger *CHIT1*⁺ cells have lost TMEM119 and some also Iba1. However, it is most likely that these larger *CHIT1*⁺ cells came from the small *CHIT1*⁺ TMEM119⁺ Iba1⁺ cells. The larger *CHIT1*⁺ CD68⁺ cells that have lost TMEM119 and/or Iba1 do not argue against the conclusion that *CHIT1* is mostly microglia-derived. As already mentioned in the discussion and the rebuttal after the first round of revisions, Iba1/*AIF1* downregulation has been reported previously in activated/disease-associated microglia. **In our new Supplementary Data 6, *AIF1* was also downregulated and one of the most important DEGs between the start and end points of the trajectories, as well as between the start and middle point (pseudotime value 7.5) of the trajectories.** As the reviewer pointed out, it is true that *AIF1* is significantly upregulated in clusters MG3, MG7, MON, CAM1 and MMC and significantly downregulated in cluster MG9 (Data file S4; now Supplementary Data 4). However, Seurat’s *FindAllMarkers* compares the gene expression of one cluster *versus* all other clusters combined and is not suited to infer a gradient in gene expression, which trajectory analysis can.

Nonetheless, we agree with the reviewer that *CHIT1*/*CHIT1* expression is not exclusively confined to microglia in transition between a homeostatic cell state and a more activated, MS-associated cell state. We do see and acknowledge that *CHIT1*⁺ cells – although lower in abundance – are also present in cluster MG5 (Fig. 4), an MS-associated microglia cluster at the end of trajectory lineage 2. At the protein level, we also observe larger *CHIT1*⁺ cells that have lost TMEM119 and/or Iba1. These cells have probably already exited the transitional cell state. To cater to this reviewer’s rightful concern, **we adjusted our interpretation throughout the manuscript and state that *CHIT1*/*CHIT1* accompanies the transition from a homeostatic towards a more activated, MS-associated cell state in microglia. In this way, we indicate that *CHIT1*/*CHIT1* expression starts around the transition, but can persist in more activated cells as well.**

A more detailed quantification of the frequencies of *CHIT1* reactivity (IBA1⁺/⁻, TMEM119⁺/⁻, CD68⁺/⁻, PLP⁺/⁻) in different lesions is missing. The used selection of markers does not help to pinpoint the *CHIT1* reactivity to the MG4 cluster and the previously mentioned discrepancies between RNA and protein levels have not been resolved. Finally, considering some of the discrepancies between the presented data and the conclusions, it is necessary to validate the specificity and reliability of the used antibodies.

Here we would like to refer to the start of our reply to comment 4). Having provided replies to the concerns above regarding antibody reliability and the transitional cell state of the *CHIT1*⁺ microglia, we do feel like we have been able to validate our most important preliminary transcriptomic findings on the protein level using the extended histology from the previous round of revisions. We have shown that both in early (rare samples that were hard to get by) and later disease stages, active MS lesions are marked by *CHIT1*⁺ microglia that start accumulating myelin debris while these were absent in non-active lesions and normal-appearing white matter. This is the strongest validation needed to provide

a biological rationale for the prognostic potential of CSF CHIT1 we identified in the first part of the paper. Moreover, our histology even suggests that CHIT1+ microglia are starting to lose homeostatic microglia markers, as also suggested by our transcriptomics, **although the exact phenotype and function of these CHIT1+ microglia should be further investigated in follow-up studies, which we now more explicitly recognize in the discussion (line 481-485).**

REVIEWERS' COMMENTS

Reviewer #3 (Remarks to the Author):

The authors have addressed my comments and revised the manuscript in a satisfactory way. The study can now be accepted for publication.